# Characterisation of an *Escherichia coli* line that completely lacks ribonucleotide reduction yields insights into the evolution of parasitism and endosymbiosis

**Samantha DM Arras[1], Nellie Sibaeva[1], Ryan J Catchpole[2†], Nobuyuki Horinouchi[3‡], Dayong Si[3§], Alannah M Rickerby[1], Kengo Deguchi[3], Makoto Hibi[3#], Koichi Tanaka[3¶], Michiki Takeuchi[3**], Jun Ogawa[3], Anthony M Poole[1]***

[1]School of Biological Sciences, University of Auckland, Auckland, New Zealand; [2]School of Biological Sciences, University of Canterbury, Christchurch, New Zealand; [3]Division of Applied Life Sciences, Graduate School of Agriculture, Kyoto University, Kyoto, Japan

**\*For correspondence:**
a.poole@auckland.ac.nz

**Present address:** †Department of Biochemistry and Molecular Biology, University of Georgia, Athens, United States; ‡Amano Enzyme Inc, Nagoya, Japan; §State Key Laboratory of Animal Nutrition, Laboratory of Feed Biotechnology, College of Animal Science and Technology, China Agricultural University, Beijing, China; #Department of Biotechnology, Biotechnology Research Center, Toyama Prefectural University, Toyama, Japan; ¶Department of Nutritional Science, Okayama Prefectural University, Okayama, Japan; **Industrial Microbiology, Graduate School of Agriculture, Kyoto University, Kyoto, Japan

**Abstract** Life requires ribonucleotide reduction for de novo synthesis of deoxyribonucleotides. As ribonucleotide reduction has on occasion been lost in parasites and endosymbionts, which are instead dependent on their host for deoxyribonucleotide synthesis, it should in principle be possible to knock this process out if growth media are supplemented with deoxyribonucleosides. We report the creation of a strain of *Escherichia coli* where all three ribonucleotide reductase operons have been deleted following introduction of a broad spectrum deoxyribonucleoside kinase from *Mycoplasma mycoides.* Our strain shows slowed but substantial growth in the presence of deoxyribonucleosides. Under limiting deoxyribonucleoside levels, we observe a distinctive filamentous cell morphology, where cells grow but do not appear to divide regularly. Finally, we examined whether our lines can adapt to limited supplies of deoxyribonucleosides, as might occur in the switch from de novo synthesis to dependence on host production during the evolution of parasitism or endosymbiosis. Over the course of an evolution experiment, we observe a 25-fold reduction in the minimum concentration of exogenous deoxyribonucleosides necessary for growth. Genome analysis reveals that several replicate lines carry mutations in *deoB* and *cdd*. deoB codes for phosphopentomutase, a key part of the deoxyriboaldolase pathway, which has been hypothesised as an alternative to ribonucleotide reduction for deoxyribonucleotide synthesis. Rather than complementing the loss of ribonucleotide reduction, our experiments reveal that mutations appear that reduce or eliminate the capacity for this pathway to catabolise deoxyribonucleotides, thus preventing their loss via central metabolism. Mutational inactivation of both *deoB* and *cdd* is also observed in a number of obligate intracellular bacteria that have lost ribonucleotide reduction. We conclude that our experiments recapitulate key evolutionary steps in the adaptation to life without ribonucleotide reduction.

## Editor's evaluation

Nearly all organisms require a ribonucleotide reductase (RNR) to convert ribonucleotides to their deoxyribonucleotide counterparts. In this important study, the reader learns how the model organism *Escherichia coli* can adapt to survive without any of its three RNRs. Compelling microbiology experiments to develop this model and analysis of compensatory mutations reveals patterns

that are conserved in the few known pathogens that have also eliminated their dependence on an RNR. The manuscript will be of interest to microbiologists, biochemists, and those who work on the evolution of microbial metabolism.

## Introduction

All life on our planet requires ribonucleotide reduction for de novo synthesis of deoxyribonucleotides, from ribonucleotides (*Nordlund and Reichard, 2006*; *Lundin et al., 2009*). The emergence of ribonucleotide reduction likely drove the transition from RNA to DNA early in the evolution of life (*Lundin et al., 2015*; *Reichard, 1993*; *Torrents et al., 2002*; *Poole et al., 2002*) and its centrality is reflected in its near ubiquity in organismal genomes (*Lundin et al., 2009*). Structural analyses indicate that the catalytic cores of ribonucleotide reductases (RNRs) are homologous (*Lundin et al., 2015*; *Sintchak et al., 2002*), despite sequence similarities between them being extremely limited (*Tauer and Benner, 1997*). RNRs catalyse deoxyribonucleotide synthesis via a common mechanism wherein a cysteinyl-free radical is generated in the active site. While this similarity in catalysis underscores a common evolutionary origin, RNRs have diverse mechanisms for radical generation (*Nordlund and Reichard, 2006*). Three broad, evolutionarily distinct, families exist which are also divided on the basis of their radical generation mechanism (*Nordlund and Reichard, 2006*; *Lundin et al., 2015*; *Lundin et al., 2010*). This divergence likely coincides with adaptation to varied environments (*Lundin et al., 2010*; *Hofer et al., 2012*). Class I enzymes are strictly aerobic in that they require oxygen or superoxide for activation (*Rose et al., 2018*). This typically, though not exclusively (*Blaesi et al., 2018*; *Srinivas et al., 2018*), occurs through oxidation of a dimetal ion centre from which a stable cysteine radical is generated, either directly, or indirectly through formation of a tyrosyl radical in the small subunit. Class III enzymes generate a stable glycine radical via cleavage of S-adenosyl methionine, and are strictly anaerobic, while class II enzymes generate a 5'-deoxyadenosyl radical through cleavage of adenosyl-cobalamin and operate irrespective of oxygen presence or absence.

To date, very few species have dispensed with ribonucleotide reduction, albeit indirectly, in that they instead rely on their hosts for deoxyribonucleotides (*Lundin et al., 2009*). Examples include a strain of *Buchnera aphidicola str. Cc* from the Cedar bark aphid, *Cinara cedri* (*Pérez-Brocal et al., 2006*). *B. aphidicola* are maternally inherited aphid endosymbionts that synthesise amino acids in short supply in the aphid diet, are obligately intracellular, and appear to be on the path to transitioning to an organelle (*Andersson, 2000*; *Tamas et al., 2002*). Two bacterial pathogens that live in very close association with their hosts—*Ureaplasma urealyticum* (*Glass et al., 2000*) and *Borrelia burgdorferi* (*Fraser et al., 1997*)—have also lost genes for ribonucleotide reduction, as have two eukaryotic parasites, *Giardia lamblia* and *Entamoeba histolytica* (*Lundin et al., 2009*).

While ribonucleotide reduction is essential, we reasoned that it should be possible to dispense with genes for ribonucleotide reduction if deoxyribonucleotides or their precursors are available via growth media. We report the successful elimination of all three RNR operons from the bacterium *Escherichia coli*. The knockout strain was created by introducing a broad-spectrum deoxyribonucleoside kinase from *Mycoplasma mycoides* (*Wang et al., 2001*), which we predicted would enable use of media-supplied deoxyribonucleosides for deoxyribonucleotide synthesis in the absence of ribonucleotide reduction. As predicted, the resulting strain is dependent on deoxyribonucleosides in the growth medium but cannot grow if the media are supplemented with deoxyribose (dR) plus the four nucleobases (A, G, C, T). To understand the impact of limited deoxyribonucleoside availability, we subjected our knockout line to experimental evolution followed by genome sequencing. We were interested in establishing whether, under such conditions, our lines would utilise the reverse deoxyriboaldolase (DERA) pathway for deoxyribonucleotide synthesis. This pathway is reversible in vitro (*Horinouchi et al., 2006a*; *Ogawa et al., 2003*), and has long been considered a plausible alternative to ribonucleotide reduction (*Racker, 1951*; *Racker, 1952*) for dNTP synthesis, with some considering it to be a plausible ancestral route for the origin of DNA (*Benner et al., 1989*; *Poole et al., 2014*). Our evolution experiments reveal that this pathway is not coopted for deoxyribonucleotide synthesis following loss of ribonucleotide reduction, with cells grown under limiting deoxyribonucleoside levels exhibiting a filamentous morphology indicative of stress. Instead, it appears that the DERA pathway is a liability under conditions of limited deoxyribonucleotide availability; we observe loss-of-function mutations in *deoB*, which are predicted to prevent catabolism of 2-deoxy-D-ribose-1-phosphate.

Available genome sequence data from obligate intracellular bacteria that lack ribonucleotide reduction indicate that *deoB* has in fact been lost on multiple occasions, suggesting that recycling of dR is disadvantageous under conditions where this sugar is in limited supply; deletion of *deoB* would enable this sugar to be rerouted for deoxyribonucleotide synthesis but would also preclude de novo synthesis via the reverse DERA reaction. Our results thus illuminate a key adaptive step taken by obligate intracellular and pathogenic species to mitigate the loss of ribonucleotide reduction.

## Results

### Creation of an *E. coli* line lacking ribonuclease reduction

*E. coli* carries genes for three RNRs: aerobic class Ia (encoded by the *nrdAB* operon), class Ib (*nrdHIEF*), and anaerobic class III (*nrdDG*) (*Figure 1*). Under aerobic growth, ribonucleotide reduction is primarily performed by the iron-dependent class Ia enzyme (*Monje-Casas et al., 2001*), while class Ib has a manganese metal centre (*Cotruvo and Stubbe, 2011*) and is therefore able to support growth when iron is scarce (*Martin and Imlay, 2011*).

We reasoned that, while ribonucleotide reduction is an essential process, it should be possible to compensate for its loss via media supplementation. To eliminate ribonucleotide reduction, we sequentially deleted the RNR operons using a scarless deletion protocol (Materials and methods). We first deleted *nrdDG*, as this operon is known not to be expressed under aerobic conditions. We next deleted the *nrdHIEF* operon from the resulting Δ*nrdDG* line. Finally, we attempted to delete the *nrdAB* operon the Δ*nrdHIEF* Δ*nrdDG* line, grown in the presence of deoxyribonucleosides (dNs). We were initially unsuccessful in deleting *nrdAB*. This is not unexpected as, while dNs are known to be taken up, some (deoxyguanine [dG], deoxycytidine [dC], deoxyadenine [dA]) are unavailable for DNA synthesis, even when genes in the DERA pathway are mutated, preventing their catabolism (*Karlström, 1970*). We reasoned that may be because of the absence of a suitable deoxynucleoside kinase activity for converting those dNs to deoxyribonucleotides. We therefore tried knocking out *nrdAB* in the presence of a heterologously expressed deoxyadenosine kinase (*dAK*) gene from *M. mycoides*. Mycoplasmas are known to possess deoxyadenosine kinases genes that permit utilisation of dNs for DNA synthesis (*Wang et al., 2001*). To establish if resulting lines lacked all three *nrd* operons, we first PCR screened for evidence of chromosomal *nrd* operon deletions. Our results indicate that all three operons were successfully deleted (*Figure 1*). As deletion does not exclude relocation of functional gene copies to another genomic location, we confirmed gene absence via PCR using primers internal to *nrd* genes and performed RT-PCR to confirm absence of gene expression (*Figure 1—figure supplement 1*). Finally, knockout status was confirmed with whole genome sequencing. This confirmed deletion of all genes for ribonucleotide reduction, with the *nrdAB* operon successfully deleted under heterologous expression of *dAK* from *M. mycoides* (*Figure 1—figure supplement 2*). Of four knockout lines, we selected one isolate (hereafter called ΔRNR) for all subsequent work. In addition to lacking all *nrd* genes, genome sequencing revealed 23 SNPs that likely appeared during the creation of ΔRNR from wild-type progenitor line (REL606) (*Supplementary file 1C*).

### ΔRNR is dependent on dN supplementation

Our ΔRNR line was created in the presence of dN supplementation, but this does not mean it is dependent on dN supplementation for growth. We therefore sought to understand what supplementation, if any, our ΔRNR line requires. To determine the lowest concentration of dNs that permits growth, we generated a series of growth curves for differing dN concentrations. At high dN concentrations (1 mg/mL), ΔRNR grows favourably, though shows a clear lag compared to wild-type on equivalent media (*Figure 2A*). At lower dN concentrations, ΔRNR growth is impaired, but there is still discernible growth at 0.05 mg/mL (*Figure 2C*). When this is dropped to 0.01 mg/mL we observed only marginal growth of ΔRNR (*Figure 2D*). As expected, in the absence of dN supplementation no growth is observed, while wild-type lines are unaffected (*Figure 2E*). Finally, we tried growing ΔRNR on dR plus the four bases (A, G, C, T). No growth was observed (*Figure 2F*). These results indicate that deletion of the three *nrd* operons from ΔRNR has completely eliminated the capacity for de novo deoxyribonucleotide synthesis; no other genes appear able to compensate for this deficiency.

We next sought to establish whether ΔRNR requires supplementation of all four dNs, or whether any are dispensable. Strains were once again grown in 1× MOPS + 1% glucose, with individual dNs

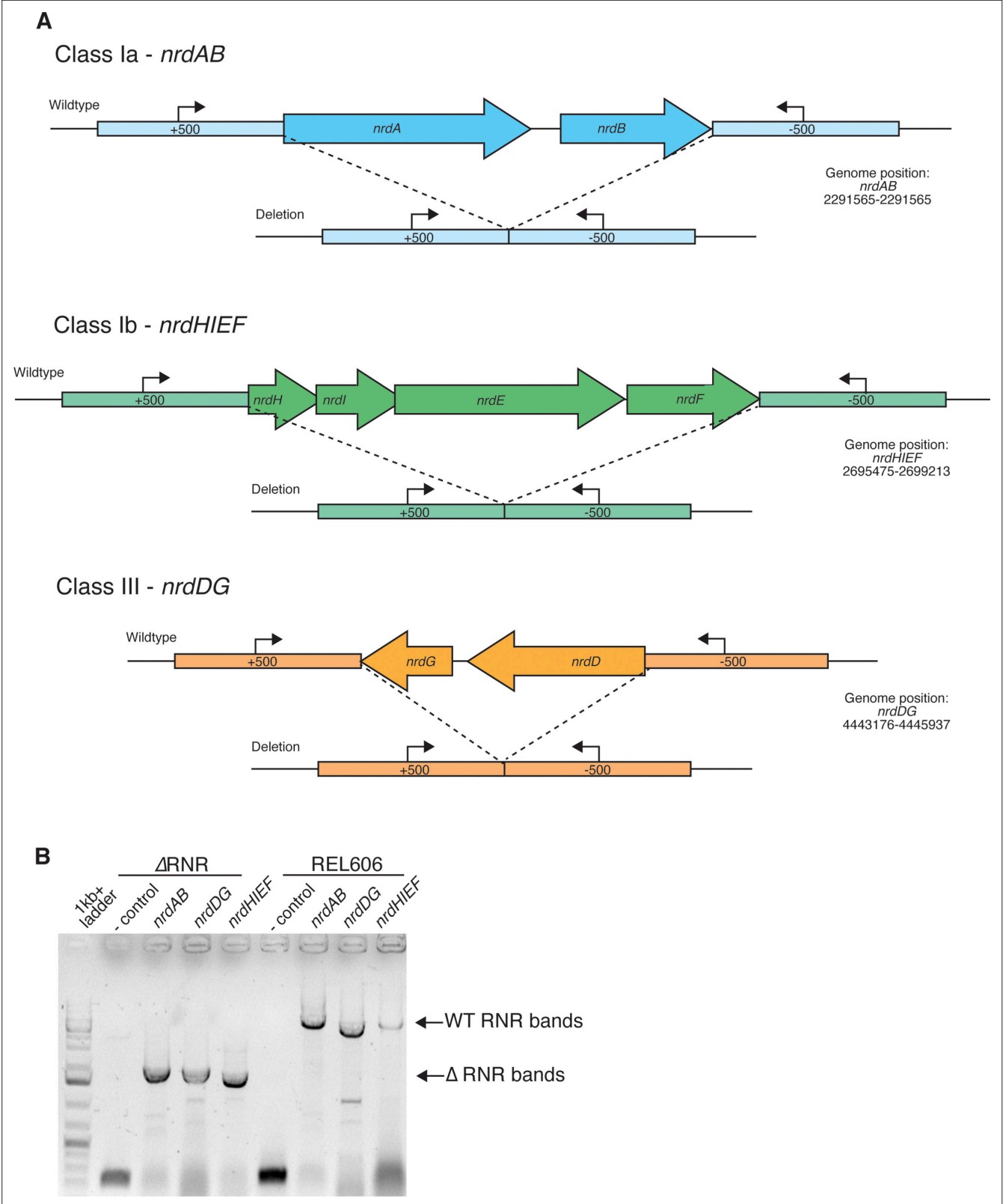

**Figure 1.** Deletion of ribonucleotide reductase (RNR) genes in *E. coli*. (**A**) Schematic of each of the three RNR operons in *E. coli* and the genomic regions following scarless deletion. Arrows indicate locations of PCR primers (***Supplementary file 1B***) used to confirm RNR operon presence/absence. (**B**) Results of PCR with primers external to all three *nrd* operons in ΔRNR compared with the wild-type progenitor strain (REL606). Lane 1, ladder; lanes 2

*Figure 1 continued on next page*

*Figure 1 continued*

and 6, no template. Lanes 3–5, band sizes consistent with operon deletion (*ΔnrdAB* 1645 bp, *ΔnrdDG* 1620 bp and *ΔnrdHIEF* 1477 bp) in the ΔRNR line. Lanes 6–9, band sizes for all three operons in wild-type *E. coli* REL606 (*nrdAB* 5303 bp, *nrdDG* 4382 bp, and *nrdHIEF* 5191 bp).

The online version of this article includes the following source data and figure supplement(s) for figure 1:

**Source data 1.** This file contains unlabelled and labelled full gel pictures for *Figure 1*.

**Figure supplement 1.** RT-PCR indicates that ribonucleotide reductase (RNR) genes are not expressed in ΔRNR.

**Figure supplement 1—source data 1.** This file contains unlabelled and labelled full gel pictures for *Figure 1—figure supplement 1*.

**Figure supplement 2.** RT-PCR indicates that Mm-dAK *is* expressed in ΔRNR.

**Figure supplement 2—source data 1.** This file contains unlabelled and labelled full gel pictures for *Figure 1—figure supplement 2*.

(dA, dG, deoxythymine [dT], or dC) added at a concentration of 0.25 mg/mL. This revealed that ΔRNR is unable to grow in the presence of dA, dG, or dT (either alone [*Figure 3A–C*] or in combination [*Figure 3E*]), even after 44 hr of monitoring (data not shown). However, ΔRNR can grow on dC alone (*Figure 3D*). Together these data indicate dC is the sole dN required for ΔRNR growth in minimal media.

## ΔRNR exhibits a filamentous cell morphology when grown under limiting dNs

During our growth assays we observed a 'clumping' phenotype when ΔRNR is grown in liquid media at concentrations of dNs that limit growth. This contrasts with the uniform cloudy appearance of wild-type *E. coli* (*Figure 4C*, top left panel). Examination of cells at ×100 magnification revealed that ΔRNR cells are elongated and filamentous at very low levels of dNs, whereas ΔRNR cells have a similar morphology to wild-type at higher dN concentrations (*Figure 4A*). Some cells reached lengths several times that of wild-type (*Supplementary file 1D*).

This phenotype appears most pronounced at low dN concentrations. We therefore sought to establish if this phenotype is the result of environmental conditions. We transferred a population of ΔRNR cells exhibiting the filamentous cell phenotype from low dN (0.01 mg/mL) media to a less restrictive environment ([dN]=1 mg/mL). Following transfer to higher [dN], the filamentous phenotype is heavily diminished (*Figure 4C*). This demonstrates that the phenotype is environmental and can be reversed if cells are moved to a high [dN] environment.

One possibility is that, under low [dN], cells are growing but unable to complete cell division. If so, this might be reflected by the presence of multiple DNA-dense regions across the length of the cells. To visualise DNA within filamentous cells, we stained ΔRNR cells grown in low [dN] (0.01 mg/mL) with FITC and DAPI. This revealed the presence of multiple DAPI-stained regions across the length of the cells (*Figure 4B*), suggestive of the presence of multiple DNA nucleoids in ΔRNR cells grown at low [dN].

## Evolution of ΔRNR lines under restricted dN supplementation

The loss of ribonucleotide reduction in obligate intracellular lifestyles presumably resulted from relaxed selection on deoxyribonucleotide production when deoxyribonucleotides are available from the environment. However, for both parasitism and endosymbiosis, the host dNTP pool must be shared. We were therefore interested to assess if the ΔRNR line adapts to a reduction in dN availability, as might occur in the evolutionary switch from de novo synthesis to dependency on host production.

In order to allow the strains to adapt to a lower concentration of dN supplementation in the growth media, we initiated evolution experiments at dN concentrations where ΔRNR lines grow, but not as well as wild-type (1 mg/mL and 0.25 mg/mL). Five independent lines of ΔRNR were serially passaged in each condition by growing the culture to stationary phase, then transferring 50 μL into a new six-well plate containing 5 mL of fresh MOPS media + 1% glucose and dNs (*Figure 5*). We also established three control lines of the wild-type progenitor strain (REL606) at each dN concentration. The initial experiment was run for a total of 30 transfers. Glycerol stocks were created every five transfers for each line, with contamination checks performed concurrently.

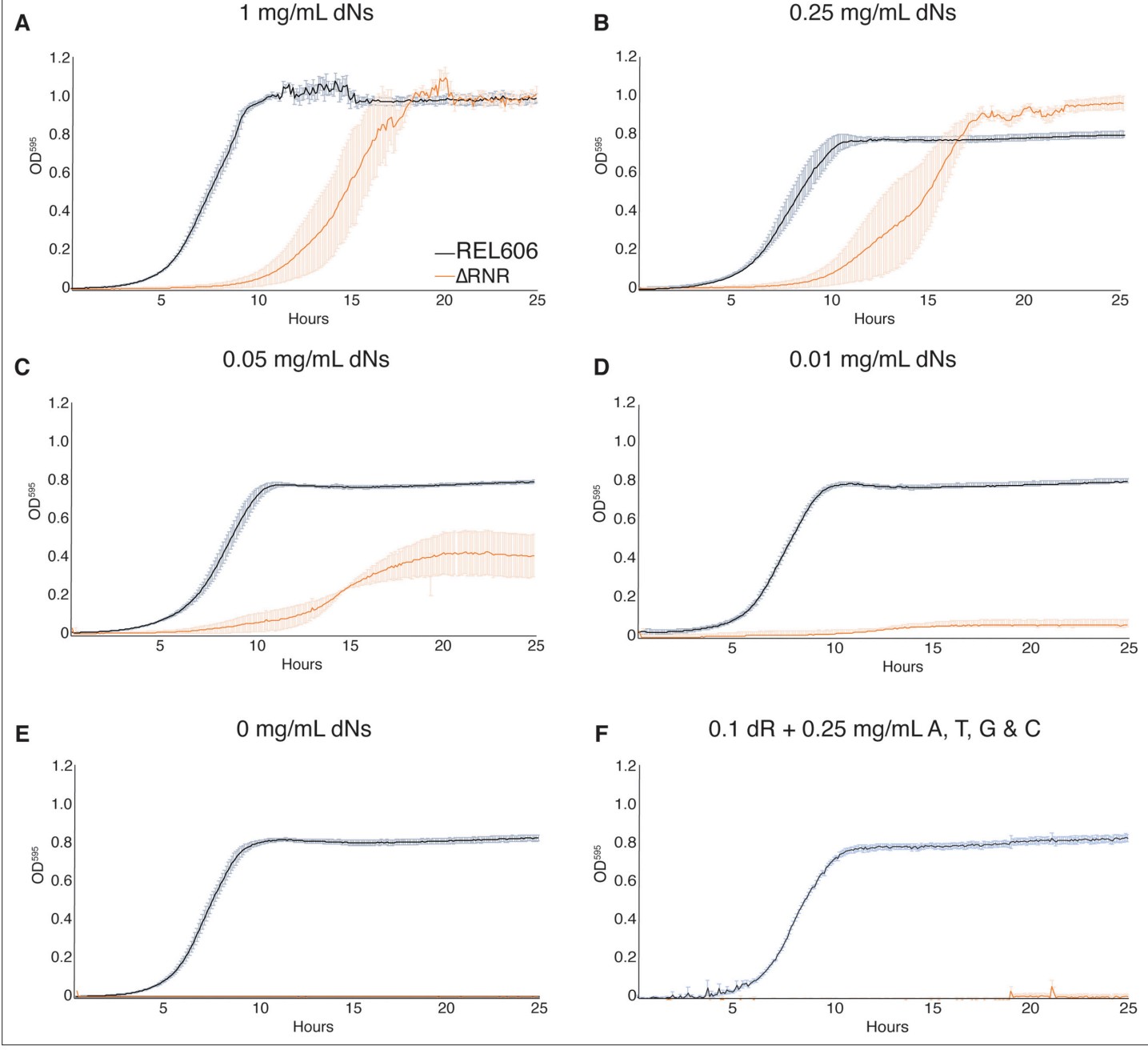

**Figure 2.** ΔRNR shows growth across a range deoxyribonucleoside (dN) concentrations but cannot grow in the absence of dN supplementation. Growth was monitored for 25 hr in 1× MOPS media + 1% glucose with supplementation, as indicated. Wild-type (REL606, black), ΔRNR (orange). Curves show mean OD$_{595}$, error bars show SEM, all experiments performed in triplicate. (**A**) 1 mg/mL, (**B**) 0.25 mg/mL, (**C**) 0.05 mg/mL, (**D**) 0.01 mg/mL, and (**E**) 0 mg/mL dNs. (**F**) 0.1 mg/mL deoxyribose (dR) + 0.25 mg/mL of adenine (**A**), cytosine (**C**), thymine (**T**), and guanine (**G**).

## Elongate cell morphology in ΔRNR lines diminishes over the course of the evolution experiment

Our initial observations of unevolved ΔRNR (henceforth ΔRNR_T0) revealed an elongated filamentous cellular phenotype at lower concentrations of dNs (*Figure 4*). We therefore monitored cell morphology during our evolution experiment. Every five transfers, we examined length of ΔRNR and REL606 cells from each condition (1 mg/mL or 0.25 mg/mL dNs; cells from well position A1 [ΔRNR] and A1 or B1 [REL606] were used for all measurements, and mean length calculated). At the beginning of the experiment, ΔRNR cells are substantially longer than wild-type, particularly at 0.25 mg/mL dNs (*Figure 6*).

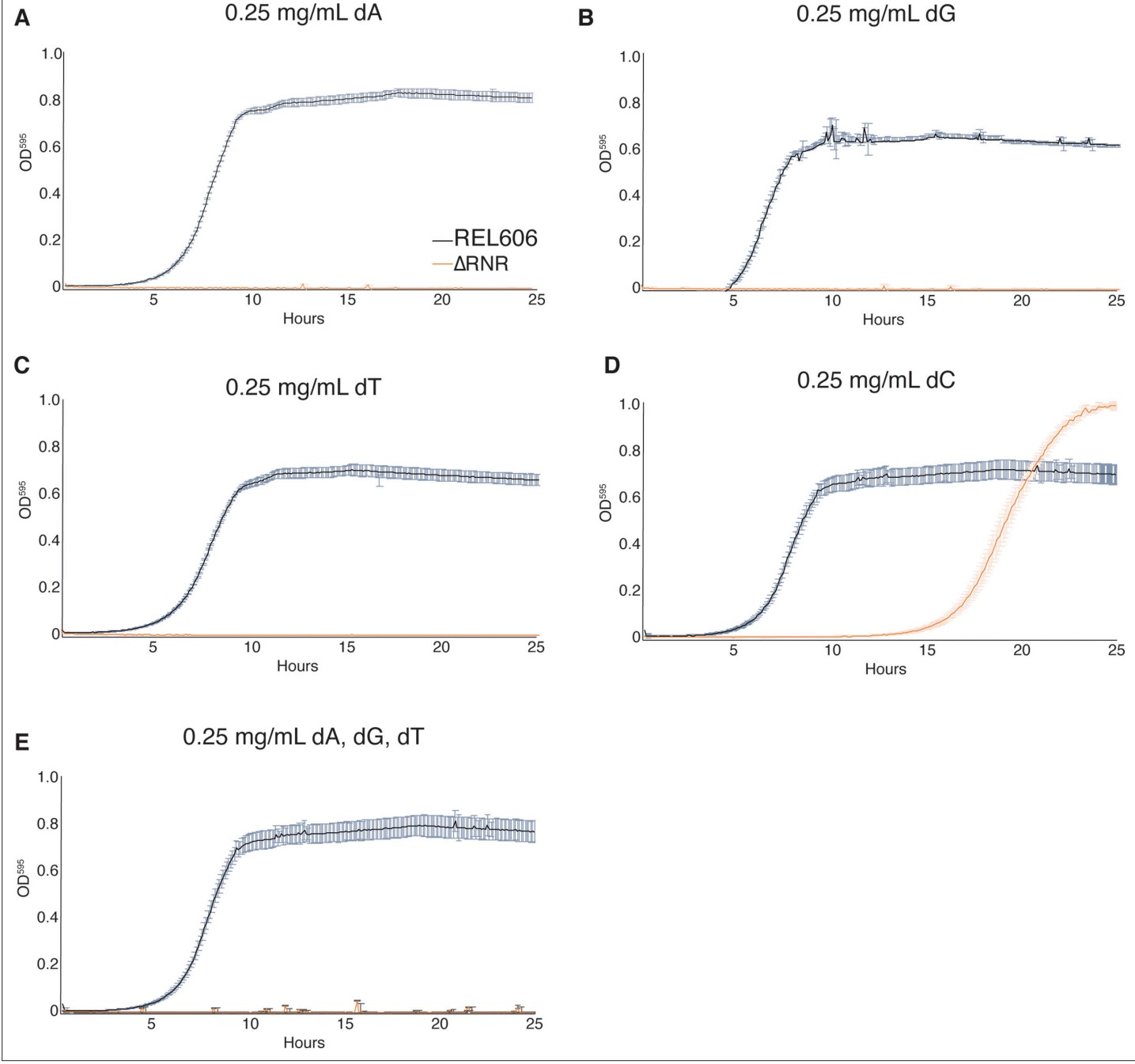

**Figure 3.** ΔRNR requires only deoxycytidine (dC) for growth in minimal media. Growth was monitored for wild-type (REL606, black) and ΔRNR (orange) in 1× MOPS media + 1% glucose, with deoxyribonucleoside (dN) supplementation, as indicated. Growth was monitored for 25 hr. Curves show mean $OD_{595}$, error bars show SEM, all experiments performed in triplicate. (**A**) Deoxyadenine (dA) (0.25 mg/mL), (**B**) deoxyguanine (dG) (0.25 mg/mL), (**C**) deoxythymine (dT) (0.25 mg/mL), (**D**) deoxycytidine (dC) (0.25 mg/mL), (**E**) dA, dG, dT, each at 0.25 mg/mL.

We recorded cell lengths of 20 µM at this lower concentration (average of 6 µM), a staggering 10–20 times the length of wild-type cells. By the time the experiment had reached 30 transfers, average cell length was comparable to wild-type at both dN concentrations (*Figure 6A*). These changes in gross morphology suggest cells had begun to adapt to restricted dN availability.

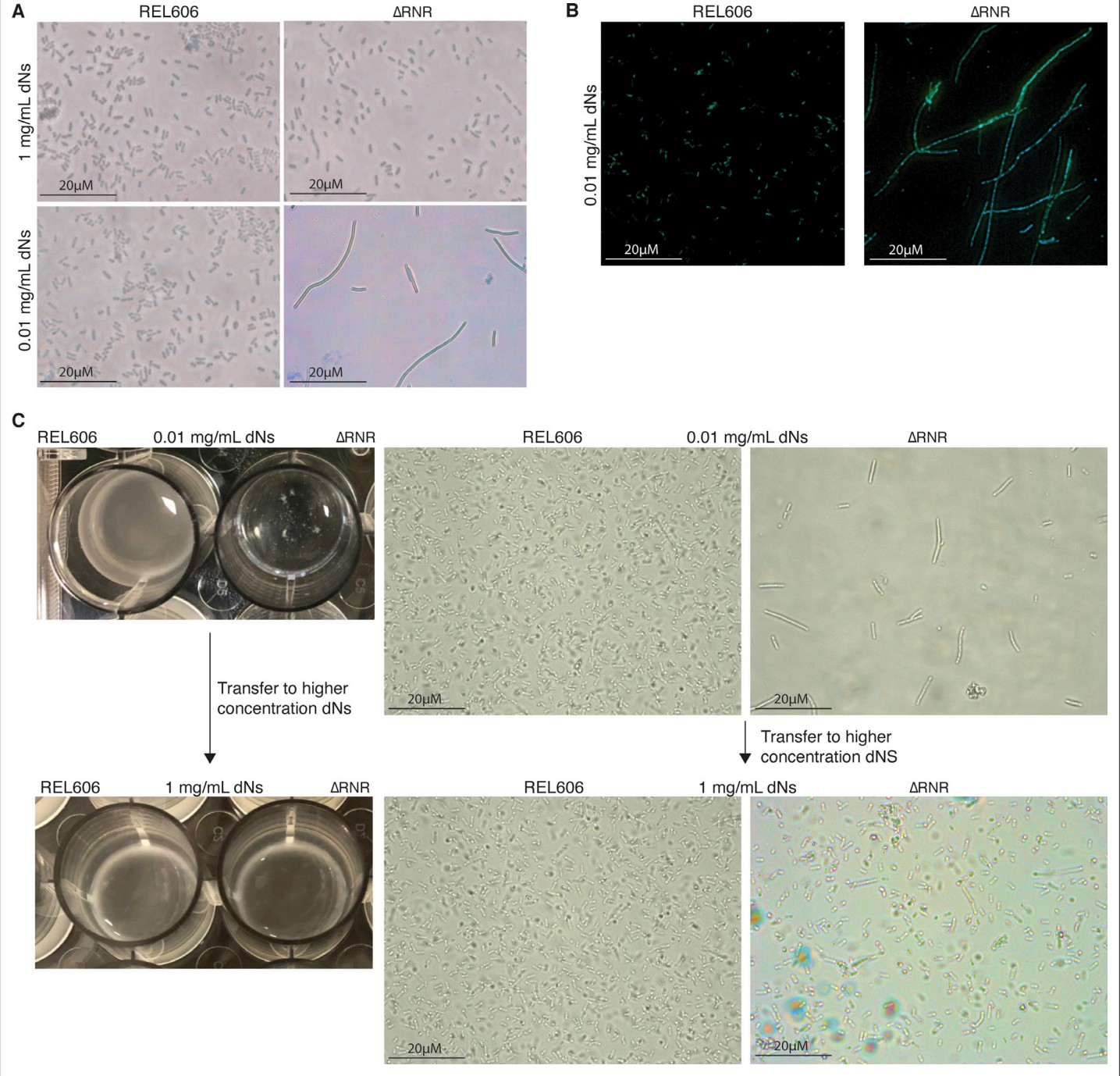

**Figure 4.** ΔRNR exhibit a reversible elongated, filamentous cell morphology with multiple nucleoids when grown in limiting deoxyribonucleoside (dN) concentrations. (**A**) ×100 Brightfield photographs of REL606 and ΔRNR at 1 and 0.01 mg/mL. (**B**) FITC and DAPI staining at ×100 of REL606 and ΔRNR at 0.01 mg/mL. (**C**) ΔRNR phenotype is reversible if dN concentration is increased. Loss of clumpy phenotype occurs when ΔRNR cells are transferred from low (0.01 mg/mL) to high (1 mg/mL) dNs. Brightfield microscopy images reveal that this change is accompanied by a reduction in elongated cell morphology upon transfer to high [dN] (magnification ×1000; scale bar 20 μM).

## ΔRNR lines exhibit improved growth following evolution under restricted dN availability

After 30 transfers, we sought to determine if ΔRNR lines had improved their capacity to grow under restricted (either 1 mg/mL or 0.25 mg/mL) dN availability. We use the following nomenclature for our evolution lines: ΔRNR_[dN] _transfer#_line#, so ΔRNR1000_T30_L1 is replicate line #1 of ΔRNR

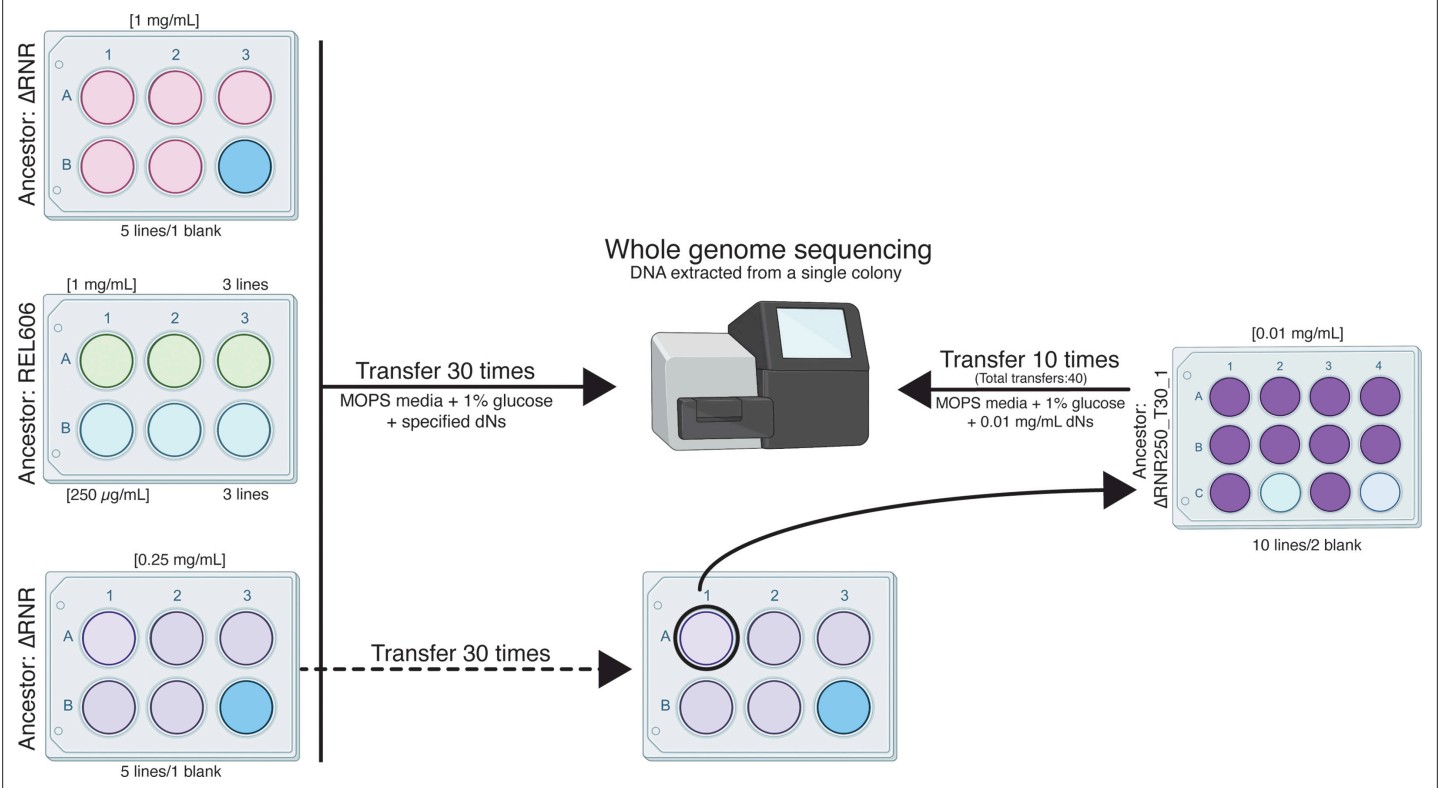

**Figure 5.** Overview of the ΔRNR evolution experiment. Five lines of ΔRNR and three lines of wild-type progenitor (REL606) were established at one of two conditions (1 mg/mL or 0.25 mg/mL deoxyribonucleosides [dNs] in MOPS + 1% glucose), and serially passaged for 30 transfers. Genomic material from each line was then extracted from a single colony and sent for sequencing giving us clonal-level genome information. To further investigate adaptation to low concentrations of dNs, the 'fittest' ΔRNR line grown at 0.25 mg/mL dNs (ΔRNR_250_T30_1) was used to seed a subsequent experiment. Ten replicate lines of ΔRNR_250_T30 _1 were serially passaged for an additional 10 transfers in MOPS + 1% glucose and 0.01 mg/mL dNs. DNA from a single colony of each of these 10 lines (ΔRNR_10_T40_1–10) was then extracted and sent for whole genome sequencing.

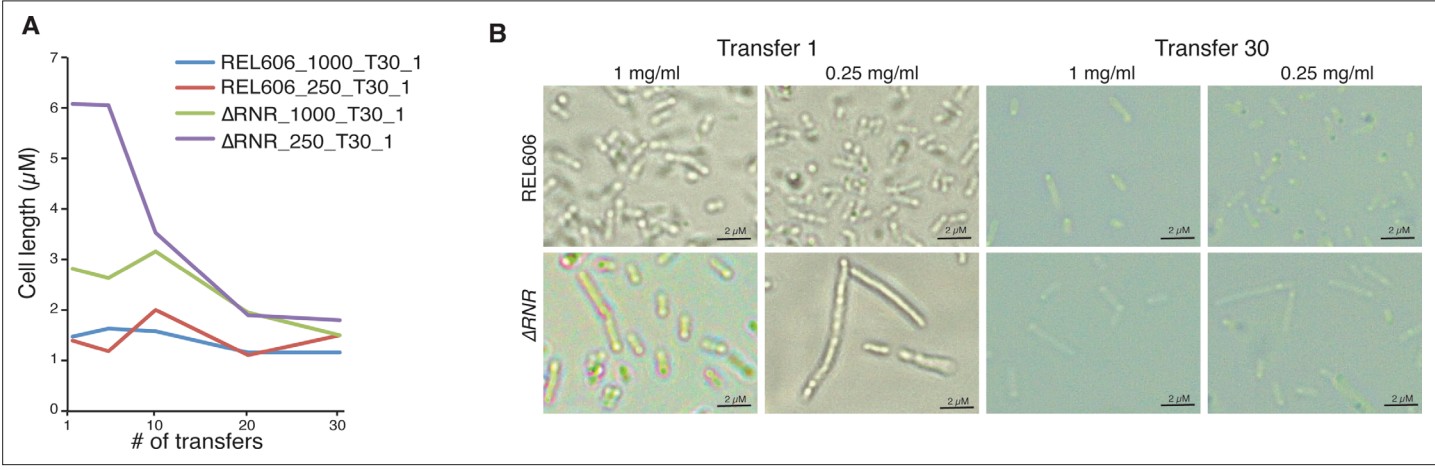

**Figure 6.** The ΔRNR elongate cell phenotype diminishes over the course of an evolution experiment. (**A**) Mean cell length during the course of a 30-transfer evolution experiment. Cell length was determined for 20 cells for replicate line 1 of ΔRNR and REL606 evolved in either 1 mg/mL or 0.25 mg/mL deoxyribonucleosides (dNs). Cell measurements were taken every five transfers. Over the course of the evolution experiment, ΔRNR cell length gradually reduces at both dN concentrations. (**B**) Morphology of wild-type (top row) and ΔRNR (bottom row) cells at transfer 1 and transfer 30 at each dN concentration (1 mg/mL or 0.25 mg/mL) (magnification ×1000; scale bar 2 μM).

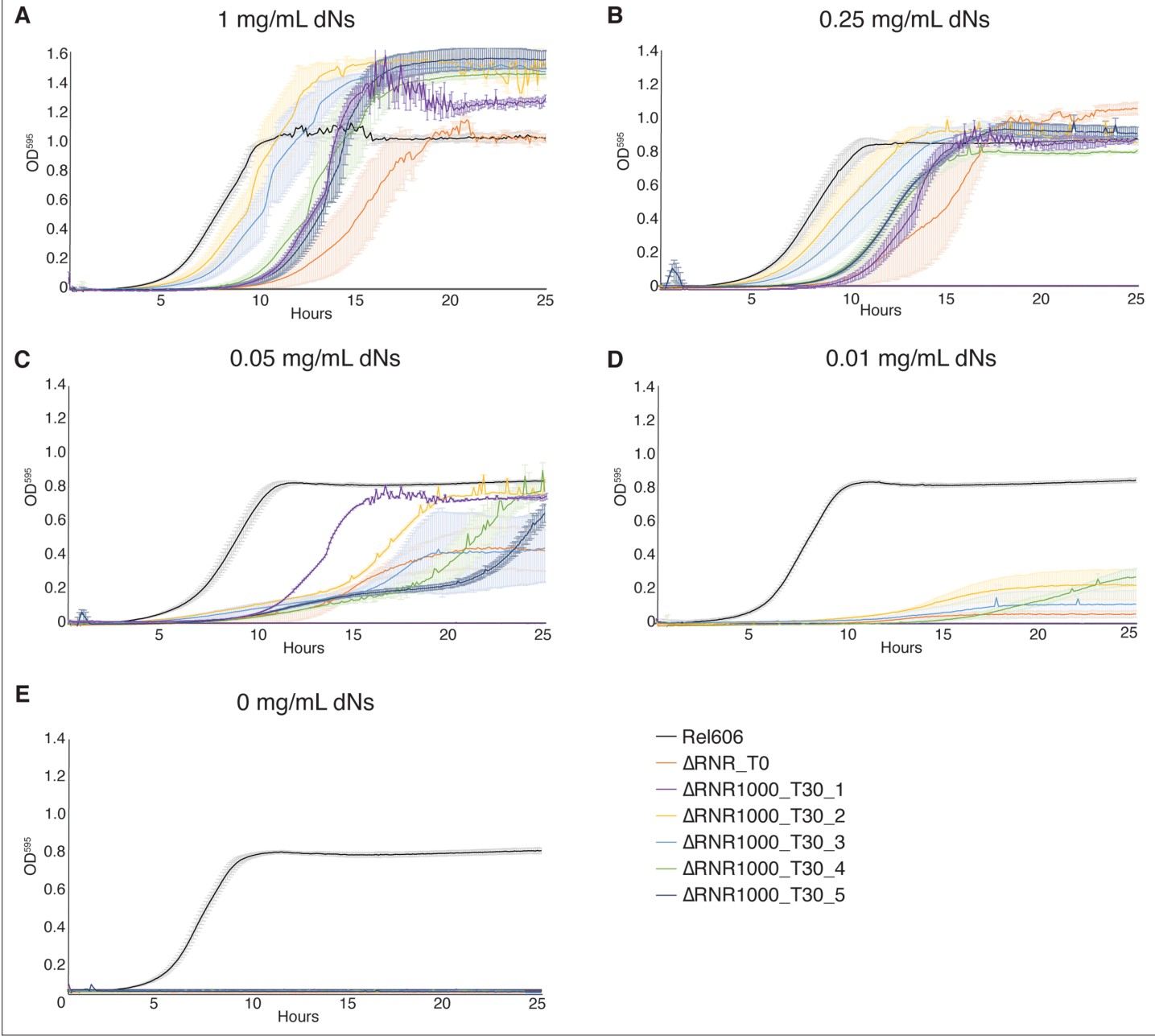

**Figure 7.** Growth characterisation of ΔRNR lines evolved at 1 mg/mL deoxyribonucleosides (dNs) for 30 transfers. Growth was monitored for wild-type (REL606), ancestor (ΔRNR_T0), and evolved lines (ΔRNR1000_T30_L1–5, evolved in 1 mg/mL dNs for 30 transfers). Growth experiments were performed in 1× MOPS media + 1% glucose with the addition of dNs at indicated concentrations. Growth was monitored for 25 hr. Curves show mean $OD_{595}$, error bars show SEM, all experiments performed in triplicate. (**A**) 1 mg/mL dNs, (**B**) 0.25 mg/mL dNs, (**C**) 0.055 mg/mL dNs, (**D**) 0.011 mg/mL dNs, (**E**) 0 mg/mL dNs.

The online version of this article includes the following figure supplement(s) for figure 7:

**Figure supplement 1.** Growth characterisation of ΔRNR lines evolved at 250 μg/mL deoxyribonucleosides (dNs) for 30 transfers.

evolved at 1 mg/mL (i.e. 1000 μg/mL) dNs for 30 transfers. At 1 mg/mL, ΔRNR1000_T30_L1 through L5 all showed improved growth relative to ΔRNR_T0 (*Figure 7A*). Moreover, compared to ΔRNR_T0, which exhibited almost no growth at 0.01 mg/mL dNs (*Figure 2D*), all evolved lines (ΔRNR1000_T30_L1–5) showed improved growth at this low concentration (*Figure 7D*). A similar overall pattern of improvement was also seen for the five ΔRNR replicate lines evolved in 0.25 mg/mL (ΔRNR250_T30_L1–5) (*Figure 7—figure supplement 1*). Note however that elongate cell morphology and clumping

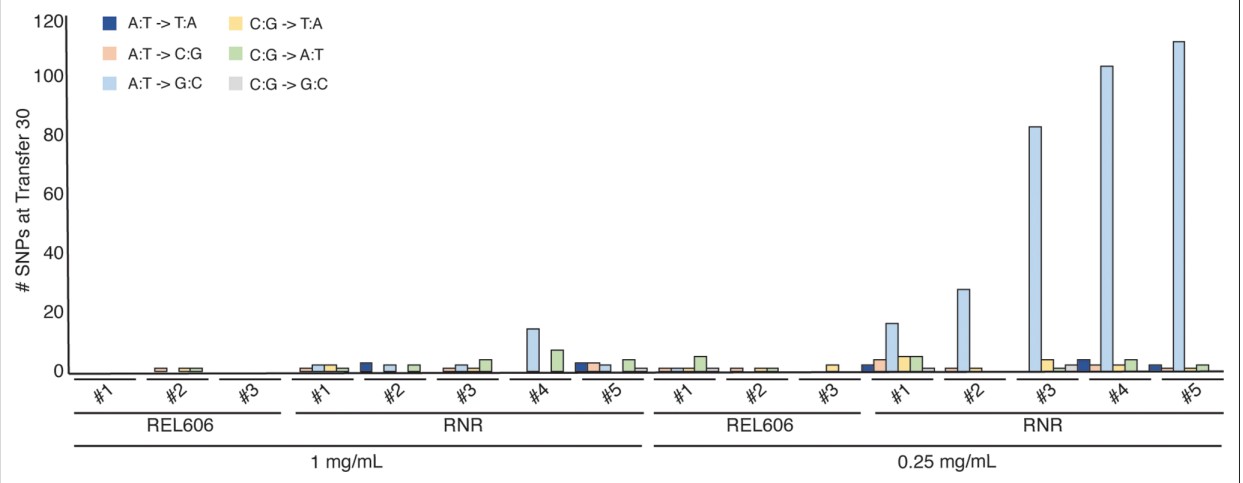

**Figure 8.** Mutations observed following experimental evolution of ΔRNR for 30 transfers. Total single nucleotide substitutions plus indels present in each line at T30. For evolution in 1 mg/mL deoxyribonucleosides (dNs), REL606=control, RNR = ΔRNR1000_T30_L1–5. For evolution in 0.25 mg/mL, REL 606=control, RNR = ΔRNR250_T30_L1–5.

The online version of this article includes the following source data and figure supplement(s) for figure 8:

**Figure supplement 1.** Mutations observed following experimental evolution of ΔRNR for 30 transfers.

**Figure supplement 2.** Schematic showing mutations detected in the *cdd* locus upon genome sequencing of all evolution lines at transfer 30 (ΔRNR_ T30_1000_L1–5, ΔRNR_T30_250_L1–5).

**Figure supplement 3.** Genomic deletions of *cdd* in transfer 30 lines are also present in the ancestor but rapidly re-evolve under the same experimental conditions.

**Figure supplement 3—source data 1.** This file contains unlabelled and labelled full gel pictures for *Figure 8—figure supplement 3*.

**Figure supplement 4.** Overview of mutational changes after experimental evolution for 10 transfers (transfers 31–40).

preclude use of OD measurements for accurate estimation of doubling time or cell counts (*Stevenson et al., 2016*) (reflected in the large standard error relative to REL606 controls), so can only be used to give a general indication of growth. All lines failed to grow in the absence of dN supplementation (*Figure 7E*, *Figure 7—figure supplement 1*), indicating they remain reliant on media supplementation.

## Evolution of ΔRNR at 0.25 mg/mL dNs results in A:T → G:C mutational skew

Following 30 transfers, we sequenced the genomes of ΔRNR_T0, our lines evolved in 0.25 mg/mL dNs (ΔRNR250_T30_L1–5) and 1 mg/mL dNs (ΔRNR1000_T30_L1–5), and each set of three wild-type control lines (REL606_1000_T30_L1–3 and REL606_250_T30_L1–3) from single colonies. The filamentous phenotype of our ΔRNR lines under restricted dN supplement precluded accurate estimation of the number of generations per transfer, which in turn precluded reliable calculation of mutation rates. We therefore report total observed mutations (single nucleotide substitutions plus indels) for each experimental line (*Figure 8*, *Supplementary file 1E*). Our evolved knockout lines (ΔRNR250_T30_ L1–5 and ΔRNR1000_T30_L1–5) all accumulated substantially more mutations than wild-type controls evolved at the same dN concentration (*Figure 8*, *Supplementary file 1E*). Our ΔRNR250_T30 lines accumulated the greatest numbers of mutations (range: 39–118 SNPs). ΔRNR1000_T30 lines did exhibit a small but significant increase in total SNPs relative to wild-type (*Figure 8—figure supplement 1* and *Supplementary file 1E*; p=0.01, unpaired t-test). There was no significant difference in total mutations between wild-type lines evolved in either 1 mg/mL or 0.25 mg/mL dNs (p=0.10; unpaired t-test).

It is well documented that RNRs keep deoxyribonucleotide pool sizes balanced through allosteric regulation (*Hofer et al., 2012*). We were therefore interested to see whether loss of ribonucleotide reduction resulted in mutational skew in *E. coli*. Our null expectation was that there would not be any detectable skew as our lines are supplemented with equal concentrations of all four dNs. If there was skew, we expected this to be most evident in our ΔRNR250_T30 lines, as these accumulated

the greatest number of mutations. Indeed, 88% of SNPs in our ΔRNR250_T30 lines are of one type: A:T→G:C (*Figure 8*, *Figure 8—figure supplement 1*).

## ΔRNR lines lose cytidine deaminase

Analysis of our genome data from lines at transfer 30 revealed that one gene, *cdd*, which codes for cytidine deaminase, was mutated in 8 of 10 lines from transfer 30 (*Supplementary file 1F and G*). Cytidine deaminase catalyses deamination of cytidine and dC to uridine and deoxyuridine. Analysis of the data revealed that seven of eight lines carried an identical deletion (*Supplementary file 1F, G*; *Figure 8—figure supplement 2*), while one line possessed an SNP that was predicted to result in a truncation. As the deletions are identical, we suspected that this deletion may have occurred in the ancestral population, but was not fixed at the time we initiated our evolution experiment. These genome data were derived from colony isolates, meaning it was not possible to determine if this was the case. Thus, the significance of this mutation was unclear in the context of our evolution experiment. To determine whether *cdd* is lost in response to the experimental conditions, we screened our original ΔRNR isolates to establish if any carried intact *cdd*. We undertook PCR screening of glycerol stocks for two knockout lines (ΔRNR31, ΔRNR34). This revealed that the glycerol stock that we used to establish our evolution experiments (ΔRNR31) was a mixed population, with both intact and *cdd* deletion present (*Figure 8—figure supplement 3*). Moreover, it appears that, after 30 transfers, all 10 lines (ΔRNR_T30_1000_L1–5, ΔRNR_T30_250_L1–5) were polymorphic for the deleted locus. Together, this indicates the deletion is very likely to have occurred in the ancestral population, and that our lines were derived from a genetically heterogeneous population, despite the knockout lines being established from individual colonies following scarless deletion of *nrdAB* (Methods and methods). Screening of ΔRNR34 glycerol stock however revealed no evidence of a deletion (*Figure 8—figure supplement 3*). We therefore established a short evolution experiment with five independent ΔRNR34 lines, in 0.25 mg/mL dNs, mimicking the early stages of our evolution experiment with the ΔRNR31 isolate. After two transfers, we observed the emergence of a *cdd* deletion in several replicate lines (*Figure 8—figure supplement 3*). In similar experiments not reported here and using these lines, we repeatedly see loss of the *cdd* locus under similar growth conditions (*data not shown*). Finally, that ΔRNR_T30_250_L5 carried an inactivating mutation in *cdd* but not the upstream locus (*yohk*), suggesting that it is deletion of the *cdd* locus that is selected under these conditions.

## ΔRNR lines evolved in 0.01 mg/mL dNs mutate the DERA salvage pathway

By transfer 30, one of our lines (ΔRNR_T30_250_L1) exhibited growth similar to wild-type when grown in 0.25 mg/mL dNs (*Figure 7—figure supplement 1*). Using this line as progenitor, we initiated a second evolution experiment using 10 parallel lines, grown in 0.01 mg/mL dNs (*Figure 5*). After 10 transfers, we sequenced these lines (ΔRNR_T40_10_L1.1–1.10). All exhibited a marked accumulation of mutations, with L1.7 accumulating 220 SNPs (*Figure 8—figure supplement 4*). As with the lines from our first 30 transfers, the majority of mutations were A:T→G:C substitutions.

Genome analysis of T40 lines revealed 648 mutations (*Supplementary file 1H*), of which 80 were found in more than one line. Of these 59 were present in two lines, 16 in three lines, and 4 in four lines (*Supplementary file 1I*).

**Table 1.** Genome sequencing of ΔRNR10_T40 lines identifies multiple mutations to *deoB*.

| Mutation | Location in ORF | Effect | Lines impacted |
|---|---|---|---|
| Substitution | 1057 (A→C) | AA substitution (T353P) | L1 L8 |
| Deletion | 583 (G(1)→G(0)) | ORF truncation (frameshift) | L2 |
| Deletion | 926 (G(4)→G(3)) | ORF truncation (frameshift) | L3 L5 L10 |
| Substitution | 932 (A→C) | AA substitution (H311P) | L4 |
| Substitution | 1147 (A→C) | AA substitution (T383P) | L6 |

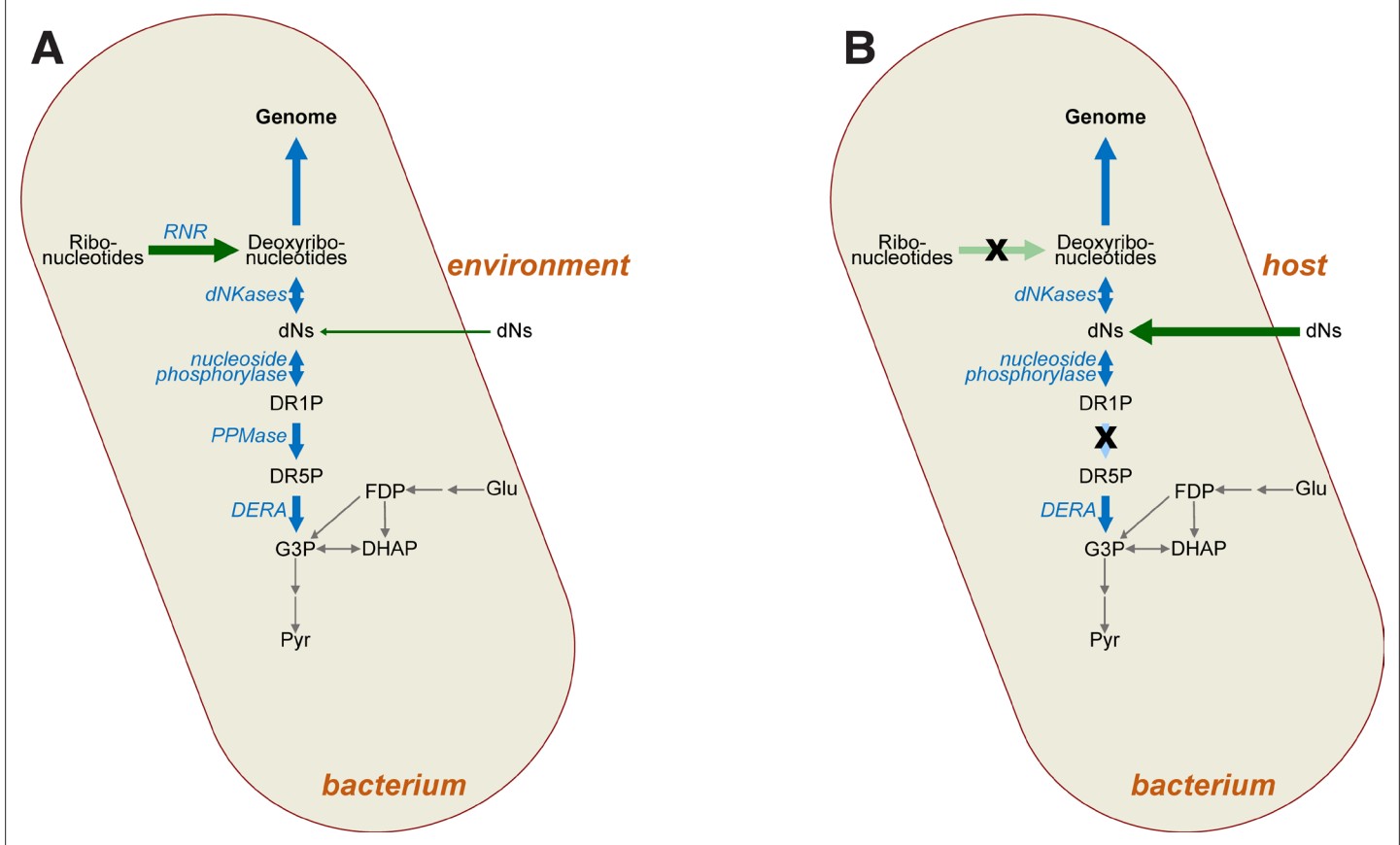

**Figure 9.** Impact of mutations on deoxyribonucleotide synthesis. (**A**) Deoxyribonucleotides in free-living bacteria are primarily synthesised intracellularly via ribonucleotide reduction (RNR), with some uptake of deoxyribonucleosides (dNs) from the environment also possible in principle. Deoxyribonucleotides are required for genome synthesis, but may also be metabolised via the DERA pathway, the end product (G3P) of which feeds into glycolysis (grey arrows). (**B**) In intracellular bacteria lacking RNR (crossed-out light green arrow), deoxyribonucleotides must instead be derived from the host as dNs (dark green arrow). Loss of RNR leads to loss of phosphopentomutase (PPMase) activity (crossed-out light blue arrow), which prevents loss of dNs via glycolysis. Abbreviations: RNR—ribonucleotide reductase; dNs—deoxyribonucleosides; DR1P—2-deoxyribose 1-phosphate; DR5P—2-deoxyribose 5-phosphate; G3P—D-glyceraldehyde 3-phosphate; DHAP—dihydroxyacetone phosphate; Glu—glucose; FDP—fructose 1,6-diphosphate; Pyr—pyruvate, dNKases—deoxynucleotide kinases; DERA—deoxyriboaldolase.

One gene, *deoB*, was mutated in eight lines (***Table 1***), and is particularly noteworthy in the context of our experiment in that it codes for phosphopentomutase. This enzyme catalyses the transfer of a phosphate group between the C1 and C5 carbon atoms of deoxribose and is responsible for the committed step in deoxyribonucleotide salvage (***Figure 9***). Closer analysis revealed that several of the mutations to *deoB* were identical. It is possible that these mutations have occurred independently, but it may also be possible that these were the result of cross-contamination between lines during the evolution experiment. To assess this, we examined whether the affected lines carried a large number of shared mutations. Lines L1.1 and L1.8 had 47 and 157 unique mutations, with only the deoB mutation shared. For this pattern to be explained by cross-contamination, this mutation would have had to have been the very first mutation to appear, and cross-contamination would have had to have occurred before any other mutation occurred. With no evidence of *deoB* mutation in the progenitor line at transfer 30 (ΔRNR250_T30_L1), we consider it unlikely that cross-contamination between these lines has occurred. In the second case, L1.3 (53 SNPs), L1.5 (40 SNPs), and L1.10 (123 SNPs) share a common *deoB* mutation along with 13 other mutations. On these data, it is harder to rule out cross-contamination. Conservatively, it thus appears that *deoB* has independently mutated five or six times under the conditions of our experiment.

To next sought to assess the impact of the mutations in *deoB*, on phosphopentomutase function. We first generated an alignment of 100 homologous phosphopentomutases, including from *E. coli* (REL606) and *Bacillus cereus* (***Supplementary file 3***). The latter was included as protein structure and

**Table 2.** Predicted impact of observed *deoB* mutations on phosphopentomutase function.

| *B. cereus* PPMase* | *E. coli* PPMase† | L1, L8 | L2 | L4 | L6 | L3, L5, L10 |
|---|---|---|---|---|---|---|
| **Active site residues** | | | | | | |
| D13 | D10 | ✓ | ✓ | ✓ | ✓ | ✓ |
| T85 | T98 | ✓ | ✓ | ✓ | ✓ | ✓ |
| D156 | D173 | ✓ | ✓ | ✓ | ✓ | ✓ |
| D286 | D306 | ✓ | - | ✓ | ✓ | ✓ |
| H291 | H311 | ✓ | - | H311P | ✓ | - |
| A327 | D347 | ✓ | - | ✓ | ✓ | - |
| H328 | H348 | ✓ | - | ✓ | ✓ | - |
| H339 | H359 | ✓ | - | ✓ | ✓ | - |
| **Non-active site residues** | | | | | | |
| I333 | T353 | T353P | - | ✓ | ✓ | - |
| T365 | T383P | ✓ | - | ✓ | T383P | - |

*PDB ID: 3M8W.
†Homologous residues derived from multiple sequence alignment (**Supplementary file 3**).

active site residues have been characterised (*Panosian et al., 2011*). Using the alignment as a guide, we next assessed functional impact of observed *deoB* mutations by mapping these onto the structure of *B. cereus* phosphopentomutase (PDB ID: C3M8). Of the five unique mutations observed in our experiment (*Table 1*), three directly impact conserved active site residues, either through truncation or mutation of active site residues (*Table 2*). The remaining two mutations are not directly associated with the active site but result in amino acid substitutions not observed in any sequences in our alignment. We therefore conclude that these mutations to the *deoB* gene are all likely to impact phosphopentomutase function, with some expected to completely inactivate the enzyme.

## Loss of ribonucleotide reduction among natural obligate intracellular bacteria is frequently associated with loss of *deoB* and *cdd* coding potential

Loss of ribonucleotide reduction genes has previously been noted in a small number species (*Lundin et al., 2009*), including the bacteria *B. burgdorferi*, *U. urealyticum*, and *B. aphidicola str. Cc*. We were therefore interested in examining if loss of *deoB* and *cdd* genes observed in our experiments extends to these species and the genera to which they belong. Sequences used for blast searches are given in *Supplementary file 1J*. For the three specific species above, blastp searches (*Supplementary file 1K*) confirmed absence of RNRs and phosphopentomutase (encoded by *deoB*). A hit to cytidine deaminase (encoded by *cdd)* was identified for *B. burgdorferi*. To examine if our results extend to other members of each genus, we repeated our searches for all members of each genus for which there are genome assemblies in the NCBI genome database (https://www.ncbi.nlm.nih.gov/genome/). For *Ureaplasma* (taxid:2129), of 58 genome assemblies, only one, *Candidatus* Ureaplasma intestinipullorum (a MAG from a chicken gut microbiome study; *Gilroy et al., 2021*) returned hits to RNR query sequences (*Supplementary file 1L*). No other *Ureaplasma* assemblies revealed evidence of RNR, PPMase, CDD, or YohK coding potential. Thus, with the possible exception of *Candidatus* Ureaplasma intestinipullorum, no members of the genus *Ureaplasma* carry genes for RNR, PPM, or CDD. These results align well with our experimental evolution study, where loss of ribonucleotide reduction resulted in subsequent loss of *deoB* and *cdd* genes.

The situation in *Borrelia* (taxid:138) is more complex. We screened 77 available genome assemblies, recovering 73 hits to NrdE and 67 to NrdF, indicating that the majority of *Borrelia* genome species code for class Ib RNR. At the time we performed our analysis, we noticed that *Borrelia burgdorferi* has been renamed *Boreliella burgdorferi*, following a proposal from *Adeolu and Gupta, 2014*,

to split *Borrelia* into two genera, *Borrelia* and *Borreliella*. While this reclassification is the subject of ongoing debate (**Barbour et al., 2017**; **Margos et al., 2017**; **Gupta and Bergström, 2019**; **Margos et al., 2018**; **Margos et al., 2020**), we expanded our search to all members of *Borreliella* (taxid: 64895, 17 species, 607 genome assemblies), and use the taxonomic distinction here so as to enable replication of our analyses. Only one member of the *Borelliella* returned any hits to RNR: *Candidatus* Borreliella tachyglossi appears to have a class Ib RNR (**Supplementary file 1M**). Given that the placement of *Candidatus* Borreliella tachyglossi is deep but appears unresolved, this may help to pinpoint the timing of RNR Ib loss from the clade/genus *Borelliella*.

For CDD, our search against *Borrelia* (taxid:138) returned 20 hits, all annotated as cytidine deaminases but these are approximately half the length of the query sequence. This is not unexpected as the *E. coli* protein is known to carry two cytidine deaminase domains and is a homodimer, whereas the shorter (homotetrameric) cytidine deaminase proteins from *Borrelia* are common across bacteria. For the same search against *Borreliella* (taxid:64895), 33 hits were returned, all annotated as cytidine deaminases. For all genome assemblies from both candidate genera, no significant hits to either PPM (*deoB*) or YohK were returned. These data indicate that no members of either *Borrelia* or *Borreliella* carry *deoB*, with only the *Borreliella* apparently lacking ribonucleotide reduction.

**Lundin et al., 2009**, noted that only one specific strain of *B. aphidicola* (from the cedar aphid *C. cedri*) lacked RNR genes. At the time, this was the smallest *Buchnera* genome, and remains one of the smallest genomes reported for a bacterium (**Pérez-Brocal et al., 2006**). A search against *B. aphidicola Str. Cc* (taxid:372461) revealed no coding potential for RNR, PPMase, CDD, and YokH (**Supplementary file 1K**). We next examined all members of *B. aphidicola* (taxid:9). This revealed widespread presence of RNR Ia (encoded by *nrdA*, *nrdB*) and PPMase (*deoB*), but no evidence of either CDD or YohK coding potential. Of 74 assemblies, we found 22 strains that lack both ribonucleotide reduction and phosphopentomutase coding, two strains that possess a <u>*deoB*</u> gene but lack genes for ribonucleotide reduction, and two with the opposite pattern (**Supplementary file 1N**).

## Discussion

While all life is dependent on ribonucleotide reduction for the synthesis of deoxyribonucleotides, a handful of species are known to lack RNR genes (**Lundin et al., 2009**) and presumably depend on their host for a source of dNTPs. In this work, we report the successful deletion of all RNR operons from *E. coli* (**Figure 1**). The resulting line (ΔRNR) was sequenced to confirm genomic deletion of RNR genes. Our ΔRNR knockout line grows on MOPS minimal media + 1% glucose supplemented with dNs (**Figure 2**), but does not grow when provided with their constituent components (dR and the four bases A, G, C, T). Growth experiments reveal that, of the four dNs, only dC is essential for growth (**Figure 3**). Under limiting levels of dNs, our lines exhibit a filamentous phenotype that can be reversed by increasing dN supplementation (**Figure 4**). Filamentous phenotypes have been documented under a range of stress conditions, including in the presence of β-lactam antibiotics (**Stevenson et al., 2016**; **Miller et al., 2004**; **Yao et al., 2012**). As cell length changes during growth in our experiment, it is not possible to use OD measurements for quantification of either doubling time or cell number (**Stevenson et al., 2016**). Our experiments should thus be taken as a qualitative indication of growth. Interestingly, we found that this phenotype reduced over the course of the experiment, suggesting that our lines were adapting to the scarcity of dNs. It will be interesting to determine whether this change is associated specifically with disruption of *deoB*, the effect of which is predicted to make dNs more readily available for DNA synthesis.

In order to understand how obligate intracellular bacteria adapted to the loss of ribonucleotide reduction, we established an evolution experiment where we serially passaged replicate lines in either 1 mg/mL or 0.25 mg/mL dNs. The resulting lines were sequenced after 30 transfers and one line evolved in 0.25 mg/mL dNs (ΔRNR_250_T30_1) was selected for a second round of evolution at 0.01 mg/mL dNs through a further 10 transfers (**Figure 5**).

Genome sequencing revealed independent mutations to two genes across our replicate lines. One gene, *cdd*, carried mutations in 8/10 lines by transfer 30. The *cdd* locus codes for cytidine deaminase, which converts dC to deoxyuridine, which in turn will be available for dTTP production. Our data reveal two independent mutations (a segmental deletion in seven lines, and an inactivating SNP in the eighth). The segmental deletion eliminated part of the *cdd* ORF and part of ORF of the upstream gene, *yohK*, which, together with *yohJ*, has been shown to code a 3-hydroxypropionate transporter

(*Nguyen-Vo et al., 2020*), and was positionally identical across all seven lines. This deletion likely occurred prior to the establishment of our replicate lines, but appeared not to have gone to fixation. An independent ΔRNR line (ΔRNR34) that lacked this deletion rapidly lost *cdd* when passaged in the experimental growth media (*Figure 8—figure supplement 3*). This, together with the independent truncation mutation in ΔRNR_T30_250_L5, which impacts *cdd* but not *yohK*, suggests there is selection against *cdd* but not necessarily *yohK*. Given the fact that our ΔRNR lines can grow with dC as sole dN supplement but do not grow if dC is omitted from the dN mix, it may be that eliminating cytidine deamination prevents loss of this key dN through deamination. We have not assessed growth on dU, but one possibility is that deoxyuridine is less amenable as a substrate for production of all four deoxyribonucleotides. One intriguing observation is that, despite the rapid loss of *cdd*, this loss of function appears not to have occurred across all replicates by transfer 30. This may be a consequence of the length of the evolution experiment. Comparing the genomes of species known to lack genes for ribonucleotide reduction (*Lundin et al., 2009*) indicates that absence of *cdd* frequently coincides with this state. Our genome analyses show that all confirmed members of the genus *Ureaplasma* lack *nrd* genes and *cdd*, suggesting that loss of *cdd* function is selectively advantageous. Establishing whether loss of one gene drives the loss of the other is not straightforward however. An analysis of strains of *B. aphidicola* revealed that 48/74 strains retain ribonucleotide reduction, but 0/74 possess *cdd* (*Supplementary file 1N*), indicating that the latter gene can be lost in lines capable of synthesising their own deoxyribonucleotides. The picture is also not clear in *Borrelia*, with *cdd* found in species variously possessing or lacking ribonucleotide reduction (*Supplementary file 1M*).

Deletion of *yohK* and *cdd* results in a small, but intact ORF (*Figure 8—figure supplement 2*). While this seems unlikely to be functional, the creation of an ORF, particularly where the upstream gene codes for a transmembrane protein, may be consistent with the 'Car Trunk' hypothesis for the avoidance of mutations that would create a toxic protein (*Omer et al., 2017*). The fact that deletion of these genes is reproducible under defined experimental conditions (*Figure 8—figure supplement 3*) suggests that this region might be suitable for testing this hypothesis.

All lines grown in 0.25 mg/mL dNs showed a marked mutational skew (A:T→G:C) that differs from the G:C→A:T mutational skew expected for *E. coli* (*Lee et al., 2012*), and more generally for bacteria (*Hershberg and Petrov, 2010*). Knockout of *cdd* does not appear linked to this skew; 4/5 lines grown at 1 mg/mL dNs do not show an obvious A:T→G:C skew (*Figure 8*). For lines grown at 0.25 mg/mL dNs, two (L2, L3) carry intact *cdd* genes and both genomes show evidence of A:T→G:C mutational skew. Such skew has been previously shown to be associated with defective mismatch repair (*Lee et al., 2012*). By transfer 30, we observe only two lines with SNPs in mismatch repair genes, with only one mutation being nonsynonymous (*Supplementary file 1*). Thus, the skew we observe cannot be explained by widespread mutation to the mismatch repair pathway. The skew may instead be driven in part by the growth dependency on dC (*Figure 3D*), with the other dNs unable to compensate for lack of dC (*Figure 3E*) but appearing to provide minor improvements to growth (*Figure 2B*). It is not currently clear why this should be so. The two major nucleoside transporters in *E. coli*, NupC and NupG, are able to transport dNs (*Patching et al., 2005*), so import does not appear to be a barrier.

We also observed loss of function of *deoB* in our experiments (*Table 1*). This observation is interesting for several reasons. First, it indicates that loss of ribonucleotide reduction does not lead to compensatory deoxyribonucleotide synthesis via the DERA pathway, which has been speculated to be an ancestral route for dNTP synthesis, predating the advent of ribonucleotide reduction (*Benner et al., 1989*; *Poole et al., 2014*). While the enzymes from this pathway can operate biosynthetically (*Horinouchi et al., 2006a*; *Ogawa et al., 2003*; *Horinouchi et al., 2006b*), loss of phosphopentomutase function via mutation of *deoB* suggests that this pathway is not readily accessible for synthesis of deoxyribonucleotides in vivo, even though there would have been a strong selective advantage to such an alternative synthetic route during our experiment. We note however that both acetaldehyde toxicity (to both DERA and cells more generally) and low intracellular availability may preclude a simple switch to using the DERA pathway for dNTP production (*Ogawa et al., 2003*; *Poole et al., 2014*).

Indeed, under the conditions of our experiment, where dN supplementation is reduced by 25-fold, we observe strong selection to dispense with *deoB*. This suggests that not only is the reverse DERA pathway inaccessible for biosynthesis, but salvage is selected against when dNTPs are scarce. In the second phase of our experiment, dN supplementation is dropped from 0.25 mg/mL to 0.01 mg/mL,

and it is under these conditions of severe dN restriction that we observe parallel loss-of function mutations at the *deoB* locus (**Table 1**). We conclude that, when deoxyribonucleotides are in short supply, *deoB* loss of function ensures that dNTPs are not lost to glycolysis via salvage (**Figure 9**). The biochemistry of the DERA pathway supports mutation of *deoB* as the most plausible target, since this step is responsible for interconversion of deoxyribose-5-phosphate and deoxyribose-1-phosphate. The latter is the substrate for phosphorylase-mediated addition of nucleobases, generating dNs, whereas the former is the substrate for DERA-mediated salvage. Thus, *deoB* loss-of-function mutations serve to prevent dR loss to central metabolism while still permitting phosphorylase-mediated dN resynthesis (**Figure 9**). Thus, while conversion of DERA to a synthetic pathway might be a better long-term solution to loss of ribonucleotide reduction, loss of function of this pathway is advantageous in the short term, precluding its cooption to deoxyribonucleotide synthesis. This interpretation is consistent with a recent experiment where mutation of dihydrofolate reductase led to *deoB* loss of function to prevent loss of deoxyribose-5-phosphate to glycolysis (**Rodrigues and Shakhnovich, 2019**).

Analysis of species that lack *nrd* genes reveals frequent loss of *deoB*, suggesting that loss of ribonucleotide reduction might drive loss of *deoB*, as observed in our experiments. For members of *Ureaplasma*, the pattern is clear (**Supplementary file 1L**): all members of this genus completely lack genes for ribonucleotide reduction and show no evidence of coding for either *deoB* or *cdd*, the two most mutated genes in our experiments. For *B.* aphidicola *cdd* appears to have been completely lost from all sequenced strains. Close to half of sequenced *Buchnera* strains show co-occurrence of ribonucleotide reduction and phosphopentomutase genes (48/74 strains), with 30% (22/74 strains) lacking genes for both functions (**Supplementary file 1N**), suggesting that loss of one function frequently results in loss of the other. However, two strains appear to carry *deoB* while lacking *nrd* genes, and two further strains show the opposite pattern. If confirmed, this would indicate loss of one function may not always result in loss of the other.

We observe a very clear pattern in members of the Borreliaceae. The NCBI Taxonomy Browser (**Schoch et al., 2020**) lists two genera with genome data, *Borrelia* and *Borreliella*, following a recent proposal (**Adeolu and Gupta, 2014**; **Barbour et al., 2017**; **Gupta and Bergström, 2019**) to split the genus *Borrelia* into two (though this proposal has been contested; **Margos et al., 2017**; **Margos et al., 2018**; **Margos et al., 2020**). Our analyses indicate that *Borrelia* (sensu **Adeolu and Gupta, 2014**) code class Ib RNRs but lack *deoB*, while *Borrelliela* lack both *deoB* and RNR genes. This pattern could be taken to indicate that members of the *Borrelia* lost *deoB* but retained ribonucleotide reduction. However, phylogenetic analyses (**Zhong et al., 2006**) and location on linear plasmids (**Miller et al., 2013**) indicate that the class Ib RNR genes (*nrdEF*) in *Borrelia* has been acquired via horizontal gene transfer, presumably following initial loss of ribonucleotide reduction in the Borreliaceae.

It is noteworthy that class Ib RNR, carried by members of the *Borrelia*, is manganese-dependent (**Cotruvo and Stubbe, 2010**), whereas class Ia is an iron-dependent enzyme (**Torrents, 2014**). Interestingly, the most well-studied member of the Borreliaceae, *Borreliella burgdorferi* (sensu Adeolu and Gupta) does not require iron (**Posey and Gherardini, 2000**). It is tempting to speculate that initial loss of ribonucleotide reduction was driven by selection for iron independence, with subsequent reacquisition of ribonucleotide reduction favouring the manganese-dependent enzyme.

More generally, the switch to an intracellular existence may result in relaxed selection on ribonucleotide reduction if host-derived deoxyribonucleotides are sufficient to replace endogenous production. This appears to be an infrequent outcome as most intracellular bacteria retain ribonucleotide reduction. This rare event might be driven by additional pressures, as evolution of iron independence in *B. burgdorferi* suggests. Our experiments indicate that, in the absence of endogenous ribonucleotide reduction, disruption of the DERA pathway via mutations to *deoB* may serve to minimise loss of these building blocks when dNTP availability is limited. Given the necessity of genome replication, we conclude that life without ribonucleotide reduction drives the loss of dNTP catabolism via DERA, enabling building block reuse for DNA synthesis.

It is interesting to note that loss of ribonucleotide reduction appears extremely rare in the evolution of endosymbiosis and parasitism, despite the seeming ubiquity of host-derived deoxyribonucleotides. Loss of production in favour of acquisition from the host ought perhaps to be a more common outcome. Acquisition from the host would appear straightforward as many bacteria are known to possess mechanisms for uptake of dNs, deoxyribonucleotides, and even DNA (**Huang et al., 2022**). With respect to dN uptake, *E. coli* for instance codes for two broad-spectrum nucleoside transporters,

NupC and NupG, which have been reported to take up dNs (*Patching et al., 2005*). These broad specificities are noteworthy in the context of our experimental results showing ΔRNR is viable when the media is supplemented with dC alone (*Figure 3*). It may be that imported dNs are not readily utilised in DNA synthesis as they must first be converted to deoxyribonucleotides for utilisation in genome synthesis. In our experiments, it was only possible to eliminate ribonucleotide reduction after first introducing a broad-spectrum deoxyadenosine kinase gene from *M. mycoides*, presumably because it improves the opportunity for imported dNs to be coopted for deoxyribonucleotide synthesis instead of being catabolised. Indeed, a recent study shows that dNs, deoxyribonucleotides, and DNA can all be used as a nitrogen source by *E. coli* (*Huang et al., 2022*). Those authors noted elevated expression of genes involved in catabolism, including *deoD*, which codes nucleoside phosphorylase, part of the DERA pathway (*Figure 9*). Thus, under conditions where DNA and its monomeric building blocks are important sources of nitrogen and phosphorus, loss of ribonucleotide reduction would presumably have a deleterious impact on growth. This trade-off may explain the small number of instances of loss of ribonucleotide reduction.

It remains to be seen whether there are conditions wherein the reverse DERA pathway (*Poole et al., 2014*) can operate synthetically in vivo as an alternative to ribonucleotide reduction, but one possibility is that this may be favoured under conditions where other nitrogen and phosphorous sources are abundant.

# Materials and methods

## Strains and growth conditions

All *E. coli* strains used in this study are listed in *Supplementary file 1A*. *E. coli* B strain REL606 (*Lenski et al., 1991*) was used as the wild-type strain for all experiments. We elected to perform experiments using REL606 as it is well suited for experimental evolution, having been subjected to a long-running evolution experiment (*Lenski et al., 1991*). Two key features that make this strain convenient for such work are genomically coded resistance against streptomycin and a selectively neutral mutation in the *araA* gene, making it easy to detect via colony colour when grown on tetrazolium arabinose. This makes it suitable for tracing during competition assays.

Strains were grown in LB (1% tryptone, 1% NaCl, and 0.5% yeast extract ± 2% agar) with the addition of 1 mg/mL each of the four dNs (dG, dA, dC, and dT (AK Scientific)) for mutant strains, unless stated otherwise. For evolution experiments, strains were grown in 1× MOPS media (*Neidhardt et al., 1974*) supplemented with dNs, as specified. All experiments were conducted in the presence of 100 µg/mL streptomycin.

## Creation of an RNR mutant

*E. coli* has three RNR operons: *nrdAB* codes for class Ia RNR, *nrdDG* codes for class III RNR, and *nrdHIEF* codes for the class Ib enzyme. Deletion of RNR operons from the BL-21 *E. coli* line REL606 (*Supplementary file 1A*) was carried out in series using a scarless genome engineering method (*Fehér, 2008*): the *nrdDG* operon (class III RNR) was deleted first, followed by *nrdHIEF* (class IB) then *nrdAB* (class IA). For each operon, 500 bp upstream and downstream from the start and stop codon was PCR amplified. A second round of overlap PCR combined fragments together and the three resulting product was cloned into pST76a. The resulting construct was transformed into wild-type at 30°C, then with genome integration induced by increasing temperature to 42°C. Helper plasmid (pSTKST) was transformed into the resulting strain, and deletion was induced with chlortetracycline. Successful deletion was confirmed using PCR across the operon (primers used in this study are listed in *Supplementary file 1B*).

To knock out the *nrdAB* operon, we sought to identify conditions where de novo deoxyribonucleotide synthesis could be replaced by supplementation. We determined that dN supplementation could replace de novo synthesis provided that we first introduced dN kinase (dNKase) activity to make dNs available for dNTP synthesis. We therefore transformed our lines with a pBAD33 construct containing a dNKase gene from *M. mycoides* (*Wang et al., 2001*) that had been previously shown to phosphorylate dA, dG, and dC. As *E. coli* codes for thymidine kinase activity (*tdk*), we reasoned that expression of the *M. mycoides* dNKase (*Mm-dAK*—characterised as a dN adenosine kinase as it has highest affinity for this dN) should render the *nrdAB* operon non-essential and thus amenable to knock out.

We cloned the *Mm-dAK* gene (codon-optimised for *E. coli* expression [GenScript; **Supplementary file 2**]) into pBAD33 under control of an IPTG-inducible *tac* promoter with *rrnB* terminator into our double knockout line (*ΔnrdDGΔnrdHIEF*). We used scarless deletion with lines grown in LB plus 0.5% wt/vol dA, dC, dG. We omitted dT as dTTP is synthesised via deamination of dCTP and dC to dUTP and dU, respectively.

The resulting strain, ΔRNR (**Supplementary file 1A**), was confirmed to lack all three operons coding for ribonucleotide reduction (Δ*nrdAB*, Δ*nrdHIEF*, Δ*nrdDG*) using PCR, RT-PCR (**Figure 1— figure supplement 1**), and whole genome sequencing.

## Total RNA extraction and RT-PCR

To confirm absence of RNR expression, strains were grown overnight in LB and diluted to 1:100 in 10 mL fresh media after reaching stationary phase. The cultures were then grown for ~3 hr and total RNA was isolated from mid-log phase cultures using TRIzol Max Bacterial Isolation kit (Thermo Fisher, Catalogue# 16096020). RNA was quantified using a Qubit 4.0 fluorometer. Purified RNA was diluted to 300 ng/mL and treated with TURBO DNaseI (Ambion, Catalogue# AM2238). This DNA-free RNA was then subjected to RT-PCR using the SuperScript III One-Step RT-PCR system with Platinum Taq DNA Polymerase kit (Invitrogen, Catalogue# 12574018) with primers specific to the gene of interest (**Supplementary file 1B**). *Mm-dAK* expression was also confirmed using RT-PCR. We observed *Mm-dAK* expression in our ΔRNR line with and without IPTG induction (**Figure 1—figure supplement 2**). We therefore elected to perform evolution experiments without IPTG.

## Growth assays

Bacterial strains were retrieved from –80°C stocks and grown overnight in LB (1 mg/mL dNs were added to media for growth of ΔRNR lines). Cultures were washed twice in 1× PBS and 10 µL equivalent was added to fresh 24-well plates containing 1 mL of 1× MOPS + 1% glucose. Differing concentrations of dNs were added from a 20 mg/mL stock solution to each individual well. Growth was monitored for 24–48 hr, taking measurements ($OD_{595}$) every 6 min using a FLUOstar Omega Microplate Reader (BMG Labtech). All experiments were performed in triplicate.

## Microscopy

Overnight cultures were grown at 37°C in MOPS media (with dNs added as required). Cultures were twice spun down and washed in 1× PBS. Ten µl of culture was aliquoted onto a microscope slide. Brightfield images were taken using a LEICA ICC50 W microscope (Leica, v.3.2.0) and imported to Photoshop (v.22.4.2) for cell length measurement. Cell length measurements were determined for each strain by averaging from 20 observations (length of the first five cells from top left to right were counted from each of four images).

To visualise DNA, we used a modified version of a FITC/DRAQ-5 double-staining protocol (**Silva et al., 2010**) instead using DAPI in place of DRAQ-5. Coverslips were coated in poly-D-lysine and placed at 37°C overnight. Coverslips were washed twice with water, dried, and stored at 4°C until ready for use. Overnight cultures were washed in 1× PBS, resuspended, and placed on a poly-D-lysine-coated coverslip at 37°C for 1 hr. Cells were washed with PBS, and fixed using 4% paraformaldehyde at room temperature for 10 min. Coverslips were washed with 1× PBS, followed by 1% Triton X-100 for 5 min. Following a further wash with 1× PBS, coverslips were then incubated in 1× PBS containing 6 µg/mL FITC for 30 min at 37°C. The coverslips were washed again with 1× PBS, then suspended in 1× PBS containing 5 µg/mL DAPI and placed in the dark for 10 min. Coverslips were lastly washed twice more in 1× PBS, then were mounted on microscope slides. Strains were visualised on a Nikon Ni-E Fluorescence microscope using fluorescent filter cubes for DAPI and FITC, and a ×100 oil objective lens. Images were overlaid using Nikon NIS-Elements software.

## Experimental evolution

All evolution experiments were performed at 37°C. Original bacterial strains were retrieved from –80°C stocks and grown overnight in LB (1 mg/mL dNs were added to media for growth of ΔRNR lines). Cultures were then washed twice in 1× PBS and 50 µL equivalent was added to fresh sixwell plates containing 5 mL 1× MOPS media (1% glucose) supplemented with the dN concentration required for each condition (1 mg/mL, 0.25 mg/mL, 0.01 mg/mL). Five lineages of ΔRNR and 3 for

WT (*E. coli* REL606; *Lenski et al., 1991*; *Jeong et al., 2009*) were passaged for each condition until transfer 30. Line 1 of ΔRNR grown at 0.25 mg/mL (ΔRNR_T30_250_1) was serially passaged as 10 independent lines for an additional 10 transfers in MOPS + 1% glucose and 0.01 mg/mL dNs (ΔRNR_T40_10_1–10). Plates were left to reach stationary with agitation at 120 rpm. Approximately $10^7$–$10^8$ of wash cells were transferred to a fresh well. A glycerol stock was created for each line every five transfers. PCR contamination checks were performed every five transfers.

## Sequencing and genome assembly

All strains and lineages required for sequencing were streaked for single colonies on LB media, with a single colony being used to inoculate an overnight culture to give clonal-level sequencing data. Genomic material was isolated using 20 μg genomic tips (QIAGEN) as per the manufacturer's instructions and DNA quantified using a Qubit 4.0 fluorometer (Thermo Fisher). Libraries were generated using the NEXTFLEX rapid XP DNA-seq kit (PerkinElmer, Catalogue# NOVA-5149-21) and NEXTFLEX UDI Barcodes (PerkinElmer, Catalogue# NOVA-514150). Strains were sequenced to approximately 50-fold coverage at Auckland Genomics Facility using an Illumina MiSeq Platform with 2×150 bp paired-end reads. Reads were trimmed using the BBDuk (*Bushnell, 2014*) 1.0 plug-in for Geneious Prime 2021.1.1 (https://www.geneious.com). Mapping to the *E. coli* REL606 genome reference was performed using Bowtie2 (v.2.3.2) (*Langmead and Salzberg, 2012*) plug-in (v.7.2.1) and visualised using default settings in Geneious.

## Sequence analyses

For creation of the phosphopentomutase alignment (*Supplementary file 3*), we first ran BLASTP with default settings (E-value threshold = 0.1, BLOSUM62, Filtering: none, Gapped: yes, Hits: 1000) against the Uniprot reference genomes plus Swiss-Prot database (https://www.uniprot.org/blast/) using the DeoB protein sequence from *E. coli* (REL606) as query. To create our alignment dataset, we filtered results for Swiss-Prot sequences and removed duplicates. We then added the *E. coli* REL606 query sequence and the phosphopentomutase from *B. cereus*, the crystal structure of which has been determined (*Panosian et al., 2011*). We generated our alignment with Muscle (*Edgar, 2004*) using the Geneious plugin (v.3.8.425) and default settings.

Three species of bacteria were noted by *Lundin et al., 2009*, as completely lacking ribonucleotide reduction: *B. burgdorferi*, *U. urealyticum*, and *B. aphidicola str. Cc*. To screen for presence/absence of ribonucleotide reduction, phosphopentomutase, and cytidine deaminases in these species, we ran blastp searches against the nr_protein database restricted to these taxa using query sequences listed in *Supplementary file 1J*. As the deletions of *cdd* involve loss of *yohK* in several instances, we also screened for presence/absence of the latter.

Searches were also performed genus/strain level in NCBI genome (https://www.ncbi.nlm.nih.gov/genome) for *Ureaplasma* (taxid:2129; 58 genome assemblies), *B. aphidicola* (taxid:9; 74 genome assemblies), and *Borrelia* (split into two genera), *Borrelia* (taxid:138, 77 genome assemblies) and *Borreliella* (taxid:64895, 17 species, 607 genome assemblies), following a proposal by *Adeolu and Gupta, 2014*. Under this proposed reclassification, *Borellia burgdorferi* has been renamed *Borreliella burgdorferi*. We note that this reclassification is the subject of ongoing debate (*Barbour et al., 2017*; *Margos et al., 2017*; *Gupta and Bergström, 2019*; *Margos et al., 2018*; *Margos et al., 2020*). As NCBI Taxonomy lists both genera and, we use the taxonomic distinction here so as to aid replication of our analyses.

Where data indicated gene absence, we confirmed this by examining the genome sequences for assembly issues, tblastn searches, and manual inspection of predicted ORFs from the genome assembly files (https://www.ncbi.nlm.nih.gov/genome/browse/#!/overview/). For cases where gene absence was clearly due to the assembly, these were excluded from further analysis. Reciprocal blast searches (blastp, nr_protein) were performed where necessary to confirm annotation of subject sequences was correct.

## Acknowledgements

We thank Auckland Genomics for technical support with genome sequencing, and the Faculty of Science Imaging Centre (University of Auckland) for image analysis.

This work was supported by a Royal Society of New Zealand Marsden Fund grant to AMP and JO (17-UOA-257).

# Additional information

## Funding

| Funder | Grant reference number | Author |
|---|---|---|
| Royal Society Te Apārangi | 17-UOA-257 | Jun Ogawa<br>Anthony M Poole |

The funders had no role in study design, data collection and interpretation, or the decision to submit the work for publication.

## Author contributions

Samantha DM Arras, Conceptualization, Data curation, Formal analysis, Investigation, Visualization, Methodology, Writing – original draft, Writing – review and editing; Nellie Sibaeva, Dayong Si, Conceptualization, Validation, Investigation; Ryan J Catchpole, Nobuyuki Horinouchi, Conceptualization, Supervision, Validation, Investigation; Alannah M Rickerby, Conceptualization, Formal analysis, Validation, Investigation; Kengo Deguchi, Investigation; Makoto Hibi, Koichi Tanaka, Michiki Takeuchi, Conceptualization, Supervision, Investigation; Jun Ogawa, Conceptualization, Resources, Supervision, Funding acquisition, Project administration, Writing – review and editing; Anthony M Poole, Conceptualization, Formal analysis, Supervision, Funding acquisition, Investigation, Visualization, Writing – original draft, Project administration, Writing – review and editing

## Author ORCIDs

Samantha DM Arras (ID) http://orcid.org/0009-0003-7544-6007
Ryan J Catchpole (ID) http://orcid.org/0000-0001-6641-647X
Jun Ogawa (ID) http://orcid.org/0000-0003-2741-621X
Anthony M Poole (ID) http://orcid.org/0000-0001-9940-2824

## Decision letter and Author response

Decision letter https://doi.org/10.7554/eLife.83845.sa1
Author response https://doi.org/10.7554/eLife.83845.sa2

# Additional files

## Supplementary files

• Supplementary file 1. Containing the following data, each in a separate sheet. (A) *E. coli* strains; (B) primers; (C) mutations in ΔRNR ancestor; (D) DRNR cells are elongated relative to wild-type; (E) T30 SNPs; (F) mutations to the cdd gene in T30 lines; (G) independent SNPs in T30 lines that impact the same locus; (H) T40 SNPs; (I) independent SNPs in T40 lines that impact the same locus; (J) query sequences for blastp searches; (K) Blastp search results confirm absence of ribonucleotide reductases (RNRs) and PPM in species listed in *Lundin et al., 2009*; (L) *Candidatus* Ureaplasma intestinipullorum hits; (M) *Candidatus* Borreliella tachyglossi hits; (N) *Buchnera* comparative analysis.

• Supplementary file 2. dAK gene from *M. mycoides* (mm-dAK) codon-optimised for *E. coli* (GenScript).

• Supplementary file 3. Alignment of phosphopentomutases.

• MDAR checklist

## Data availability

All genome data have been deposited to the Single Read Archive (https://www.ncbi.nlm.nih.gov/sra) under accessions SRX17677859-SRX17677886. The following biosamples are associated with this project: SAMN30957267-SAMN30957273. The data have been deposited with links to BioProject accession number PRJNA882995 in the NCBI BioProject database (https://www.ncbi.nlm.nih.gov/bioproject/).

The following dataset was generated:

| Author(s) | Year | Dataset title | Dataset URL | Database and Identifier |
|---|---|---|---|---|
| Arras SDM, Sibaeva N, Catchpole RJ, Horinouchi N, Si D, Rickerby AM, Deguchi K, Hibi M, Tanaka K, Takeuchi M, Ogawa J, Poole AM | 2022 | Experimental evolution of an *Escherichia coli* line lacking ribonucleotide reduction | https://www.ncbi.nlm.nih.gov/bioproject/?term=PRJNA882995 | NCBI BioProject, PRJNA882995 |

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
