## [Editor Report]

Nearly all organisms require a ribonucleotide reductase (RNR) to convert ribonucleotides to their deoxyribonucleotide counterparts. In this important study, the reader learns how the model organism *Escherichia coli* can adapt to survive without any of its three RNRs. Compelling microbiology experiments to develop this model and analysis of compensatory mutations reveals patterns that are conserved in the few known pathogens that have also eliminated their dependence on an RNR. The manuscript will be of interest to microbiologists, biochemists, and those who work on the evolution of microbial metabolism.

---

## [Decision Letter]

**Decision letter after peer review:**

Thank you for submitting your article "Characterisation of an *Escherichia coli* line that completely lacks ribonucleotide reduction yields insights into the evolution of obligate intracellularity." for consideration by *eLife*. Your article has been reviewed by 3 peer reviewers, one of whom is a member of our Board of Reviewing Editors, and the evaluation has been overseen by Christian Landry as the Senior Editor. The following individual involved in the review of your submission has agreed to reveal their identity: Anders Hofer (Reviewer #2).

Essential revisions:

1) Revise the introductory text about cofactor usage by class I RNRs recommended by reviewer #1. Also, modify the discussion of pathogens as detailed by reviewer #2 and comment on the selection of the starting strain as suggested by reviewer #3.

2) Make the changes to Figure 9 that are described by reviewer #2.

3) Comment further on how observation of cell elongation would impact OD measurements (reviewer #1 comment) and address the concerns of reviewer #3 about cell morphology changes and analysis of evolutionary trajectory.

4) Modify the abstract and introduction to include a statement about the need to introduce a nucleotide kinase gene.

5) Describe the control experiment to identify selective pressures induced by the antibiotic.

6) Clarify whether population-level sequencing experiments were performed as detailed by reviewer #3.

7) Revise Figure 1 to include the entire gel and include more effective annotations.

*Reviewer #1 (Recommendations for the authors):*

Line 72-73 – revise terminology referring to "an oxygen-containing metal ion centre" to describe Cys radical generation by class I RNRs. The Cys oxidant cofactor employed by class I enzymes does not contain oxygen. Class I RNR cofactors are assembled from either oxygen or an oxygen-derived oxidant (superoxide). But the important part about the oxygen dependence during assembly is not the incorporation of oxygen per se – it has not been established whether that happens in every case anyway – but the fact that oxygen provides oxidizing equivalents that can convert the precursor form of the cofactor (Mn(II), Fe(II), etc) to an oxidant with sufficient potency to oxidize a Cys side chain. Also, class Ie RNRs do not contain metal in their activated form – so it is also not accurate to say that all class I RNRs use a metal center to accomplish Cys oxidation.

Line 284 – there are a number of more recent studies that support the idea that class Ib RNR is used under conditions of iron limitation in *E. coli*.

Line 643 – missing word.

*Reviewer #2 (Recommendations for the authors):*

The broad questions are already described in the Public Review section (major findings and possible future experiments). Generally, I think the findings are very interesting but I have a few things that I would like to be corrected.

I find it a bit misleading that it is mentioned in the manuscript that the few organisms that lack RNR are obligately intracellular and Borrelia burgdorferi is mentioned as one of them. B. burgdorferi is mainly extracellular in the mammalian host, although it has been suggested that it can have intracellular stages. Another extracellular pathogen that lacks RNR is Giardia (several species). This is a protozoan parasite (eukaryote) that is extracellular. I think this organism should be mentioned as well in the manuscript. I am also surprised that it is stated that Mycoplasma is known to be dependent on salvage synthesis for dNTP production (row 300). They actually encode a novel subclass of class I RNRs that was published a few years ago (DOPA radical-dependent RNRs). Remove the sentence or explain this part further. The statement of lacking RNR is valid for Ureaplasma but not Mycoplasma.

I also think some adjustments need to be made in Figure 9. First of all, I think it would be good to use "ribonucleotides" and "deoxyribonucleotides" instead of NTPs and dNTPs in the top part of the figure showing the RNR reaction. This will both solve the problem that many RNRs use diphosphates as substrates and also that dNKs phosphorylate deoxyribonucleosides to the monophosphate level and not the triphosphate level. Another problem with the figure is that PPase normally means pyrophosphatase and because of that I think it is better to write "nucleoside phosphorylase" or alternatively just "phosphorylase" if the space is limited. A third problem is that it looks like the cells are taking up dNTPs, which is usually not the case, at least not in prokaryotes that do not have endocytosis. The uptake arrow from the outside should instead lead to dNs.

It should be stated in the abstract and introduction that they needed to introduce an exogenous deoxynucleoside kinase gene to the *E. coli* cells in order to get the system to work (*E. coli* can normally only phosphorylate thymidine). I would also like them to discuss more what is known about the substrate specificity of the kinase and what transporters *E. coli* has for the uptake of the deoxynucleosides.

*Reviewer #3 (Recommendations for the authors):*

Specific questions/comments for the authors:

– A general question, why did the authors choose REL606 as opposed to other strains? Is this due to evolutionary experiment convenience? If so, please briefly discuss this.

– Please clarify if a clonal and/or population-level sequence was performed, and how.

– What is the control experiment (i.e. a strain evolved in parallel) to detect any selective pressures that could be introduced by the strep in the evolution media?

– What is the significance of the cell morphology analyses for the evolutionary trajectory hypotheses presented in the study? An earlier justification or a better-justified discussion would be helpful here. To my knowledge, REL606 does not particularly exhibit a morphologically distinct response to many environments the bug has been tested/evolved before, or is this not the case? If so, then why is this a key observation attributable to the hypothesis tested here?

– Is there a correlation between the number of mutations vs. the morphological modifications of the evolved cell?

– Please clarify what is meant by this sentence (line 414): Note however that elongate cell morphology and clumping preclude use of OD measurements for accurate estimation of doubling time or cell counts (reflected in the large standard error relative to REL606 controls), so can only be used to give a general indication of growth.

– I'm intrigued by the cross-contamination speculation (and lack thereof) for the T30.L1 line. Did the authors perform population-level sequencing? Shouldn't that reveal whether early contamination took place?

– As a general comment, the second half of the Results section is rather difficult to follow. Some arguments are emphasized in this part (especially the loss of deoB and cdd genes section) and others can move to Discussion; there is a lot of back/forthing going on in this section, making it harder to grasp the sequence of events. Similarly, some paragraphs emphasize results are in the Discussion section (see, for instance, section starting line 653).

– Please include what blue/red lines correspond to on Figures 2 and 3 directly, in addition to the legend.

– Figure 1 should include the entire gel, not a trimmed version. Include an arrow pointing to exactly what we are looking at here.

– Overall summary figure (indicating the "Sequence" would be a good place to specify what you mean by sequence: and how (clonal, genome, how many replicates, etc.)).

– Specify what you mean by length (uM) on the figure in addition to the caption. The colors here do not indicate the specific strain on the Figure – is red the WT? Blue evolved WT? Unclear.

---

## [Author Response]

Essential revisions:1) Revise the introductory text about cofactor usage by class I RNRs recommended by reviewer #1. Also, modify the discussion of pathogens as detailed by reviewer #2 and comment on the selection of the starting strain as suggested by reviewer #3.

We have now reworded the introductory text for accuracy and added references to recent papers reporting on the advances that reviewer #1 notes (ll 72-76). We also added references regarding the recent studies showing a likely role for Ib in replication during iron starvation (ll 295-296). Thanks for alerting us to these.

Thanks to Reviewer #2 for noting some inaccuracies in our descriptions of the key lineages that do not have RNRs. We have corrected these errors. We had originally omitted the eukaryote cases as our work is focused on bacteria, but agree it is helpful to mention these, so we have now added these in for completeness (ll 81-89).

We have also corrected the statement regarding the presence of ribonucleotide reductase in *Mycoplasma* (l 312). We now cite the discovery of DOPA radical-dependent RNRs in the introductory text. (ll 73-74)

Regarding the choice of starting strain, we did indeed choose this because of some of the convenient features of this strain for experimental evolution, including genomically coded antibiotic resistance and presence of a neutral SNP in a gene involved in arabinose utilisation, which makes this lineage suitable for competition experiments. We made use of the former in the project, but not the latter.

We have now made a statement regarding choice of strain under the methods subsection, ‘Strains and growth conditions’. (ll 125-131)

2) Make the changes to Figure 9 that are described by reviewer #2.

Figure 9 has been updated for clarity, following advice from Reviewer #2. We have replaced NTPs/dNTPs with ribonucleotides and deoxyribonucleotides, respectively. Thanks for this suggestion. While we and others have used the abbreviation PPase elsewhere, we do appreciate the point, and have altered this to remove any ambiguity. PPase is now replaced with “nucleoside phosphorylase". The arrow from outside the cell has been corrected to lead to the deoxyribonucleotides label inside the cell, and the cell boundary has been redrawn for clarity. We have also updated the figure legend (ll 834-845).

3) Comment further on how observation of cell elongation would impact OD measurements (reviewer #1 comment) and address the concerns of reviewer #3 about cell morphology changes and analysis of evolutionary trajectory.

We mentioned this in the abstract, but we have now added brief mention of this in the final paragraph of the introduction, where we summarise key findings (ll 109-111).

For establishing if there is either growth or no growth under various conditions, as we have done, a qualitative assessment such as the one presented in Figure 3 is sufficient. The issue of whether OD is impacted by cell elongation has been documented previously (Stevenson et al. https://www.nature.com/articles/srep38828), and is only a problem if trying to quantify parameters such as doubling time or when trying to estimate cell counts. We do not do either of these, as calculation of both requires an assumption of normal cell morphology in *E. coli*. We have added a note to clarify this in the first paragraph of the Discussion section, as per the suggestion from Reviewer #1. (ll 606-613)

4) Modify the abstract and introduction to include a statement about the need to introduce a nucleotide kinase gene.

We have now added statements to this effect in the abstract (ll 38-39) and in the final paragraph of the introduction (ll 94-98).

5) Describe the control experiment to identify selective pressures induced by the antibiotic.

Both the control and the experimental lines are resistant to streptomycin. The basis of the resistance is a genomic mutation in the *rpsL* gene which occurred following selection for resistance in the creation of REL606 (see Studier et al. 2009; https://doi.org/10.1016/j.jmb.2009.09.021). As streptomycin is included in the growth media of both the control and experimental lines, it should not affect interpretation of results.

If additional mutations do appear as a result of antibiotic resistance or some other form of media adaptation, they would be expected to appear in both control and experimental lines as both are evolved in the same antibiotic-containing growth media. Our genome sequencing indicated only one gene is mutated in both control and experimental lines. This gene, *kdbD*, encodes a putative histidine kinase. Across the control and experimental lines, we observe a range of mutations, including synonymous and nonsynonymous substitutions, plus one apparent frameshift in *kdbD*. Mutation of this gene cannot be specifically linked to any selection pressures associated with antibiotic resistance, and we note that this gene is not well studied. However, we plan to revisit this matter in a follow-up paper where we plan to present detailed population-level genomic data (see response to point 6, below).

Given that antibiotic presence is a constant across all lines, and given that we do not see evidence of specific adaptation to this antibiotic (as the lines are already preadapted to its presence) we do not believe it is necessary to discuss this, so we have not made any changes to the manuscript.

6) Clarify whether population-level sequencing experiments were performed as detailed by reviewer #3.

As noted in the Methods section (under Sequencing and genome assembly), genome data are derived from single colonies (ll 235-237). We have since performed population-level sequencing which we will present as part of a subsequent study, where we have extended the evolution experiment through to 100 transfers, and examined population dynamics.

7) Revise Figure 1 to include the entire gel and include more effective annotations.

Figure now edited and figure legend updated.

Reviewer #1 (Recommendations for the authors):Line 643 – missing word.

Corrected, thanks.

Reviewer #3 (Recommendations for the authors):– Please clarify if a clonal and/or population-level sequence was performed, and how.

Clonal – see above.

– What is the control experiment (i.e. a strain evolved in parallel) to detect any selective pressures that could be introduced by the strep in the evolution media?

Note that a control experiment is presented, but not for Streptomycin, for reasons noted above.

– What is the significance of the cell morphology analyses for the evolutionary trajectory hypotheses presented in the study? An earlier justification or a better-justified discussion would be helpful here. To my knowledge, REL606 does not particularly exhibit a morphologically distinct response to many environments the bug has been tested/evolved before, or is this not the case? If so, then why is this a key observation attributable to the hypothesis tested here?

See above. The cell morphology is reversible, so it is not significant in evolutionary terms. As now explained in the revised text, this type of morphology has been seen in a range of conditions, and is associated with stress. We show it is specifically a response to the environment (i.e. low [dN]) and is alleviated when cells are grown in higher [dN]. Perhaps what is interesting is that this environmental phenotype appears to be less severe as the experiment proceeds. This might be due to improved capacity to utilise dNs, perhaps via reduction of loss to catabolism following mutation of deoB. We now make this suggestion in the discussion (l l613-617).

– Is there a correlation between the number of mutations vs. the morphological modifications of the evolved cell?

No. We show that the morphology is environmental, not genetic – our cell lines lose this phenotype when grown in higher [dNs], though note above comment about mutations that might influence the availability of dNs for DNA synthesis.

– Please clarify what is meant by this sentence (line 414): Note however that elongate cell morphology and clumping preclude use of OD measurements for accurate estimation of doubling time or cell counts (reflected in the large standard error relative to REL606 controls), so can only be used to give a general indication of growth.

This is now discussed, as noted above. We cite a key paper (Stevenson et al.) which explains why these types of cellular morphology/behaviour preclude use of OD measurements for estimating doubling time/cell counts.

– I'm intrigued by the cross-contamination speculation (and lack thereof) for the T30.L1 line. Did the authors perform population-level sequencing? Shouldn't that reveal whether early contamination took place?

We have not performed population sequencing for the present work. However, we are currently undertaking this for a follow-up study which takes our evolution experiment through to 100 transfers. As the reviewer notes, with population data, we should indeed be able to confirm the status of the SNP in deoB in lines 1 and 8, and we look forward to presenting that work in our next article.

– As a general comment, the second half of the Results section is rather difficult to follow. Some arguments are emphasized in this part (especially the loss of deoB and cdd genes section) and others can move to Discussion; there is a lot of back/forthing going on in this section, making it harder to grasp the sequence of events. Similarly, some paragraphs emphasize results are in the Discussion section (see, for instance, section starting line 653).

We presume that, by the ‘second half’ of the results, the reviewer is referring to the bioinformatics-based analyses presented. We presented the results of screens for several genes for three genera, and elected to present results by genus, rather than by gene. We believe this is the more logical way to follow for understanding the results in biological context, but it does inevitably result in some back and forth, since we are discussing the same gene sets in several genera.

Regarding the results from line 653, we felt it useful to reemphasise the key results, but we have now added some further discussion around uptake of deoxyribonucleosides, which we hope puts this in clearer context. This is at ll 680-682 in the revised manuscript.

– Please include what blue/red lines correspond to on Figures 2 and 3 directly, in addition to the legend.

Corrected.

– Figure 1 should include the entire gel, not a trimmed version. Include an arrow pointing to exactly what we are looking at here.

Corrected.

– Overall summary figure (indicating the "Sequence" would be a good place to specify what you mean by sequence: and how (clonal, genome, how many replicates, etc.)).

Corrected. The legend to Figure 5 now reads:

“Five lines of ∆RNR and three lines of wild type progenitor (REL606) were established at one of two conditions (1 mg/mL or 0.25 mg/mL dNs in MOPS+1% glucose), and serially passaged for 30 transfers. Genomic material from each line was then extracted from a single colony and sent for sequencing giving us clonal level genome information. To further investigate adaptation to low concentrations of dNs, the “fittest” ∆RNR line grown at 0.25 mg/mL dNs (∆RNR_250_T30_1) was used to seed a subsequent experiment. Ten replicate lines of ∆RNR_250_T30 _1 were serially passaged for an additional 10 transfers in MOPS+1% glucose and 0.01 mg/ml dNs. DNA from a single colony of each of these 10 lines (∆RNR_10_T40_1-10) was then extracted and sent for whole genome sequencing”.

We have also indicated this in methods and in the main text, and we have also altered the text in the figure itself to make it clear we sequenced from single colonies.

– Specify what you mean by length (uM) on the figure in addition to the caption. The colors here do not indicate the specific strain on the Figure – is red the WT? Blue evolved WT? Unclear.

Corrected. Figure 6 has been updated to include the axis label to read “cell length (uM)”. The missing colour indicators in the key have been updated.